# Scavenger receptor-A is a biomarker and effector of rheumatoid arthritis: A large-scale multicenter study

Fanlei Hu [1,2,14✉], Xiang Jiang[1,3,14], Chunqing Guo[4,5,6,14], Yingni Li[1], Shixian Chen[7,8], Wei Zhang[9], Yan Du[10], Ping Wang[1], Xi Zheng[1,3], Xiangyu Fang[1], Xin Li[1,3], Jing Song[1,3], Yang Xie[1], Fei Huang[1], Jimeng Xue[1], Mingxin Bai[1], Yuan Jia[1], Xu Liu[1], Limin Ren[1], Xiaoying Zhang[1], Jianping Guo[1], Hudan Pan [11], Yin Su[1], Huanfa Yi[4,12], Hua Ye[1], Daming Zuo [4,13], Juan Li[7,8], Huaxiang Wu[10], Yongfu Wang[9], Ru Li[1], Liang Liu[11✉], Xiang-Yang Wang [4,5,6✉] & Zhanguo Li [1,2,3✉]

Early diagnosis is critical to improve outcomes in rheumatoid arthritis (RA), but current diagnostic tools have limited sensitivity. Here we report a large-scale multicenter study involving training and validation cohorts of 3,262 participants. We show that serum levels of soluble scavenger receptor-A (sSR-A) are increased in patients with RA and correlate positively with clinical and immunological features of the disease. This discriminatory capacity of sSR-A is clinically valuable and complements the diagnosis for early stage and seronegative RA. sSR-A also has 15.97% prevalence in undifferentiated arthritis patients. Furthermore, administration of SR-A accelerates the onset of experimental arthritis in mice, whereas inhibition of SR-A ameliorates the disease pathogenesis. Together, these data identify sSR-A as a potential biomarker in diagnosis of RA, and targeting SR-A might be a therapeutic strategy.

[1] Department of Rheumatology and Immunology, Peking University People's Hospital & Beijing Key Laboratory for Rheumatism Mechanism and Immune Diagnosis (BZ0135), Beijing, China. [2] State Key Laboratory of Natural and Biomimetic Drugs, School of Pharmaceutical Sciences, Peking University, Beijing, China. [3] Peking-Tsinghua Center for Life Sciences, Peking University, Beijing, China. [4] Department of Human & Molecular Genetics, Virginia Commonwealth University, School of Medicine, Richmond, USA. [5] Institute of Molecular Medicine, Virginia Commonwealth University, School of Medicine, Richmond, USA. [6] Massey Cancer Center, Virginia Commonwealth University, School of Medicine, Richmond, USA. [7] Department of Traditional Chinese Internal Medicine, School of Traditional Chinese Medicine, Southern Medical University, Guangzhou, China. [8] Department of Rheumatology, Nanfang Hospital, Southern Medical University, Guangzhou, China. [9] Department of Rheumatology and Immunology, First Hospital Affiliated to Baotou Medical College & Inner Mongolia Key Laboratory of Autoimmunity, Baotou, China. [10] Department of Rheumatology, the Second Affiliated Hospital, Zhejiang University School of Medicine, Hangzhou, China. [11] State Key Laboratory of Quality Research in Chinese Medicine, Macau University of Science and Technology, Macau, China. [12] Present address: Central laboratory of Eastern Division, The First Hospital of Jilin University, Changchun, China. [13] Present address: Department of Immunology, School of Basic Medical Sciences, Southern Medical University, Guangzhou, China. [14] These authors contributed equally: Fanlei Hu, Xiang Jiang, Chunqing Guo. ✉email: fanleihu@bjmu.edu.cn; lliu@must.edu.mo; Xiang-Yang.Wang@vcuhealth.org; li99@bjmu.edu.cn

Rheumatoid arthritis (RA) is a chronic autoimmune disease that can lead to joint destruction, disability, and premature mortality[1-3]. An estimated 50% of RA patients become permanently work disabled within 2–3 years of diagnosis[4-6]. It affects about 1% of the population worldwide and a large-scale survey of residents showed that the prevalence of RA in China is 0.28%[7]. Although early diagnosis of RA are often associated with better response to treatment, reduced co-morbidity, and lower mortality, the rate of disease remission is only 8.6%[4,8].

The current RA classification criteria proposed by ACR/EULAR in 2010 improves sensitivity for early detection of RA as compared to the former classification system proposed by ACR in 1987[9,10]. The biomarkers, i.e., rheumatoid factor (RF) and anti-cyclic citrullinated peptide antibody (anti-CCP) used in the current classification criteria only show a modest discriminating power. The sensitivity and specificity are 67% and 95% for anti-CCP, and 69% and 85% for RF, respectively[11]. Although several biomarkers have been identified in peripheral blood or synovial fluid of RA patients[12], none of those achieves better specificity and sensitivity than anti-CCP alone. Therefore, there remain unmet needs to develop additional diagnostic tools as well as treatment options for RA.

Scavenger receptor-A (SR-A), also termed CD204, is a pattern recognition receptor primarily expressed on the cells of myeloid origin and displays pleiotropic biological functions[13]. SR-A was initially identified as a major receptor on macrophages for internalization of modified lipoproteins[14]. A wealth of studies have described the roles of SR-A in lipid metabolism, cardiovascular diseases, and pathogen recognition[15-17]. Our previous research has established an immunoregulatory activity of SR-A in attenuating cancer vaccination-induced antitumor immune responses[18-22] and in limiting T-cell activation in inflammatory hepatitis[23,24]. Both cell-associated SR-A (cSR-A) and soluble SR-A (sSR-A) exhibit T cell suppressive activity via functional regulation of innate immune cells, e.g., dendritic cells (DCs) and myeloid-derived suppressor cells[23,24]. However, the significance of sSR-A as a potential diagnostic marker in RA has not been investigated.

In the present study, we perform a large-scale, multicenter study to assess the discriminative power of serum sSR-A for RA diagnosis. An elevation of sSR-A is present exclusively in patients with RA, but not in patients with systemic lupus erythematosus (SLE), Sjogren's Syndrome (SS), osteoarthritis (OA), ankylosing spondylitis (AS), gout, psoriatic arthritis (PsA), ANCA-associated vasculitis (AAV), adult onset still's disease (AOSD), polymyalgia rheumatica (PMR), autoimmune hepatitis (AIH) or non-autoimmune inflammatory diseases (NAID). sSR-A not only demonstrates potential discriminatory ability for RA, but also complements the diagnosis for early RA as well as for seronegative RA. Using mouse arthritis models, we provide compelling evidence to show that increasing the levels of SR-A accelerates arthritis progression whereas inhibition of SR-A ameliorates disease severity. These findings support diverse functions of SR-A under different pathological conditions, and might provide diagnostic and therapeutic strategies for clinical management of RA.

## Results

### The prevalence of sSR-A in RA.
We first assessed the levels of sSR-A in patients with RA or other types of rheumatic diseases and non-autoimmune inflammatory diseases. As shown in Fig. 1a, the serum levels of sSR-A in patients with RA were significantly higher than those of healthy individuals or patients with other common rheumatic diseases, including SLE, SS, OA, AS, Gout, PsA as well as AAV, AOSD, and PMR. Although elevation of sSR-A was previously reported in patients with hepatitis[23,24], the levels of sSR-A in RA patients were much higher compared to those in patients with autoimmune hepatitis (AIH, Fig. 1b). In addition, sSR-A was elevated in RA patients but not in patients with non-autoimmune inflammatory diseases (NAID), including enteritis, gastritis, pneumonia, and colitis (Fig. 1c). Strikingly, SLE or SS patients complicated with RA showed fundamentally increased levels of sSR-A compared with those without RA (Fig. 1d). All these results suggested that sSR-A was selectively elevated in RA patients.

### The diagnostic value of sSR-A in RA.
To determine the potential diagnostic value of serum sSR-A for RA, we performed a large-scale, multicenter study. A total of 2616 serum samples were collected, including 1454 samples in the Beijing cohort (training cohort), 740 and 422 samples in the Inner Mongolia and Hangzhou cohort, respectively (validation cohorts 1 and 2). We showed that the levels of sSR-A in RA patients within the training cohort were substantially higher than those of patients with other rheumatic diseases and healthy controls (median 2.77 ng/mL, mean 9.04 ng/mL, SD 14.24 ng/mL, $p < 0.001$, Fig. 2a). The same results were also seen in the two validation cohorts (Fig. 2b, c), which was further confirmed by the pooled data from the three cohorts (Fig. 2d).

The covariate-adjusted receiver operating characteristic curve (AROC) analysis using non-parametric method[25-27] was performed to evaluate the performance of sSR-A in diagnosis of RA, with age and gender as the confounders identified by the multivariable logistic regression analysis. The result revealed a significant area under the age-adjusted and gender-adjusted ROC curve (AAUC) of 0.8420 (95% CI extending from 0.8094 to 0.8688) for sSR-A in Beijing cohort (Fig. 3a). The AAUCs of Inner Mongolia and Hangzhou cohorts were 0.8641 (95% CI extending from 0.8232 to 0.9013) and 0.8219 (95% CI extending from 0.7502 to 0.8748), respectively (Fig. 3b, c). Pooling the data of the three cohorts yielded an AAUC of 0.8436 for sSR-A (Fig. 3d), approximate with that for anti-CCP (0.84) and RF (0.83) as reported[28]. These results indicate that sSR-A reveals potential capacity in distinguishing RA.

The optimal cut-off value in the study was set for 3 SD above the mean value of the healthy controls, which showed better clinical utility of sensitivity and specificity than the ROC curve and Youden index analysis. Based on the threshold value of 1.7024 ng/mL, the sensitivity and specificity of sSR-A for identification of RA were 61.36% and 94.38% in Beijing cohort, 73.24% and 90.51% in Inner Mongolia cohort, 74.19% and 83.15% in Hangzhou cohort, respectively. Analysis of the pooled three cohorts revealed the sensitivity of 66.41% and specificity of 91.45% for sSR-A in RA diagnosis, with PPV of 80.19% and NPV of 83.94% (Table 1). All these suggested that sSR-A demonstrated a diagnostic value slightly lower than anti-CCP, but higher than RF. The reported average sensitivity and specificity are 67% and 95% for anti-CCP, and 69% and 85% for RF, respectively[11].

The performance of sSR-A was then compared with ESR and CRP, the two indexes listed in ACR/EULAR 2010 classification criteria. RA patients from the training and validation cohorts as well as the pooled cohort were divided into the following four groups, and the levels of sSR-A as well as the positive rates were further analyzed: RA patients with normal ESR and normal CRP, RA patients with normal ESR and increased CRP, RA patients with increased ESR and normal CRP, RA patients with increased ESR and increased CRP. The results showed that sSR-A demonstrated elevated levels with high prevalence in all these four groups. Even in RA patients with normal ESR and normal CRP, the positive rate of sSR-A still reached 57.58% (57/99) in the pooled three cohorts (Fig. 4a–d). All these results indicate that sSR-A provides a complementary value to ESR and CRP.

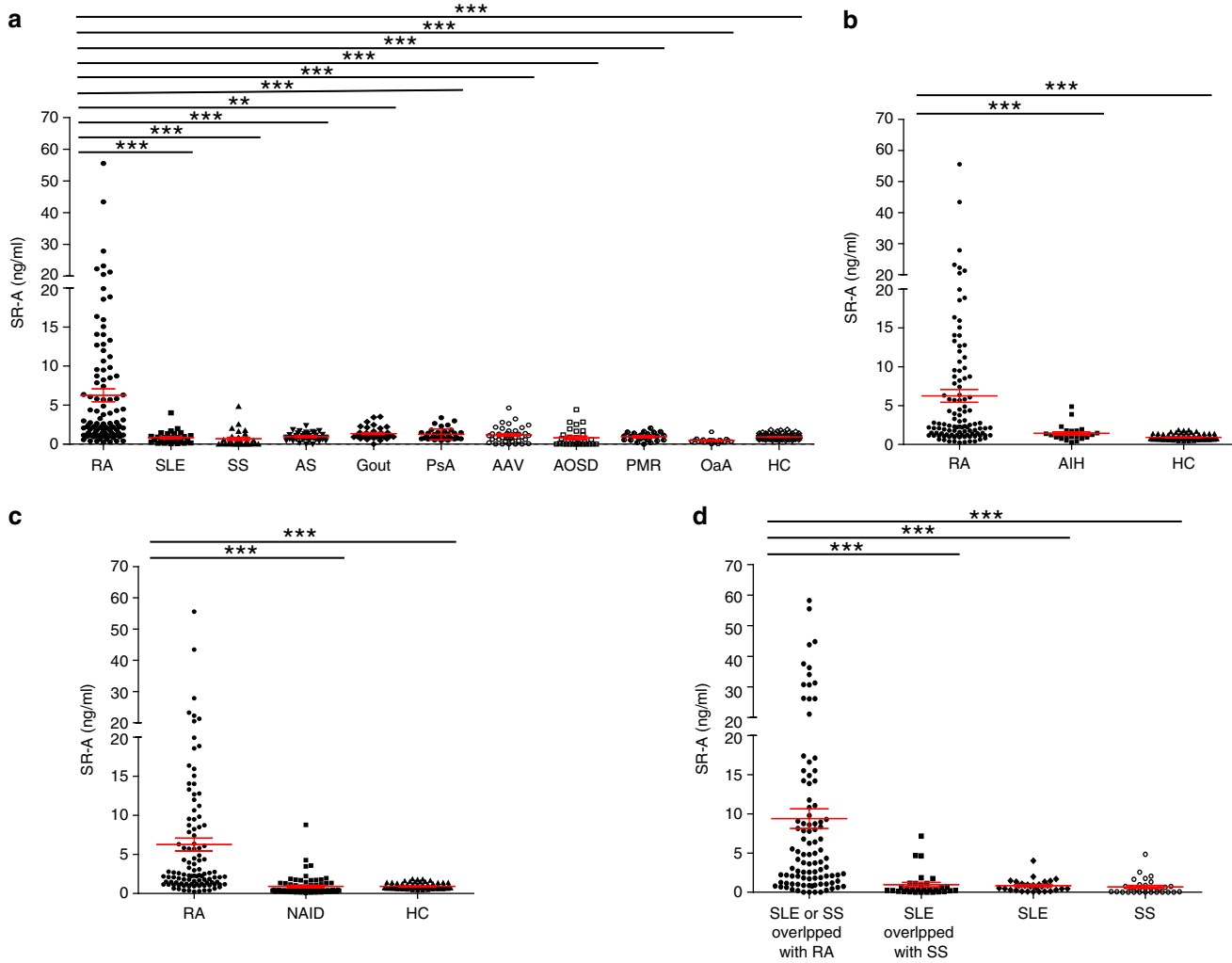

**Fig. 1 The prevalence of sSR-A in patients with RA. a–c** The serum levels of sSR-A were significantly higher in patients with RA than those of healthy individuals and patients with other common rheumatic diseases and non-autoimmune inflammatory diseases. **a** patients with other common rheumatic diseases (systemic lupus erythematosus (SLE, ***p = 1.1346E−10), Sjogren's Syndrome (SS, ***p < 1E−15), ankylosing spondylitis (AS, ***p = 1.1289E−08), Gout (**p = 0.0035), psoriatic arthritis (PsA, ***p = 1.3426E−04), ANCA-associated vasculitis (AAV, ***p = 2.0262E−07), adult onset still's disease (AOSD, ***p = 2.98E−13), polymyalgia rheumatic (PMR, ***p = 8.2141E−06), and osteoarthritis (OA, ***p = 2E−15)) and healthy controls (***p = 4E−15); **b** patients with autoimmune hepatitis (AIH, ***p = 7.8003E−04) and healthy controls (***p < 1E−15); **c** patients with non-autoimmune inflammatory diseases (NAID, including enteritis, gastritis, pneumonia, and colitis, ***p < 1E−15) and healthy controls (***p = 3.8E−14). **d** sSR-A were higher in SLE or SS overlapped with RA patients than SLE overlapped with SS patients (***p = 9.8292E−10), SLE patients (***p = 3.1014E−07) and SS patients (***p = 1.9579E −11). RA = 107; SLE = 30; SS = 30; AS = 39; Gout = 39; PsA = 39; AAV = 39; AOSD = 30; PMR = 24; OA = 20; HC = 90; AIH = 25; NAID = 71; SLE or SS overlapped with RA = 98; SLE overlapped with SS = 31. Red horizontal lines: means; error bars: SEMs. **p < 0.01, ***p < 0.001 (Kruskal–Wallis test followed by Dunn's posttest for multiple comparisons). Source data are provided as a Source Data file.

Early diagnosis and timely treatment of RA are critical for reducing co-morbidity and mortality. We thus examined the diagnostic value of sSR-A in early stage RA (ERA). In the three training and validation cohorts, there were 167 ERA patients in total with disease duration <24 months. To more accurately elucidate the prevalence of sSR-A, we collected additional 84 ERA patient serum samples from Beijing cohort. In these 251 early RA patients, sSR-A showed substantial diagnostic value. In early RA patients with disease duration <12 and <24 months, the positive rates of sSR-A were 60.11% (107/178) and 63.35% (159/ 251), respectively. In early RA patients with disease duration <6 months, sSR-A also showed a 53.85% (49/91) prevalence (Fig. 5a).

Since anti-CCP and RF are routinely used for RA diagnosis, we assessed the complementary diagnostic value of sSR-A in patients

lacking these two disease-specific antibodies. Serum samples from 179 anti-CCP-negative RA patients, 276 RF-negative RA patients, and 155 (anti-CCP and RF)-double negative RA patients (including samples from the three training and validation cohorts) were subjected to the detection of sSR-A levels and prevalence. The positive rates of sSR-A in anti-CCP-negative and RF-negative RA patients were 49.72% (89/179) and 39.13% (108/ 276), respectively. More importantly, in (anti-CCP and RF)- double negative RA patients, sSR-A also demonstrated a 42.58% (66/155) prevalence (Fig. 5b). These results support the use of sSR-A to faciliate the diagnosis in anti-CCP and/or RF-negative RA patients.

The value of sSR-A as a predictor was also examined. Serum samples from 119 undifferentiated arthritis (UA) patients were further collected for sSR-A detection. Although lower than

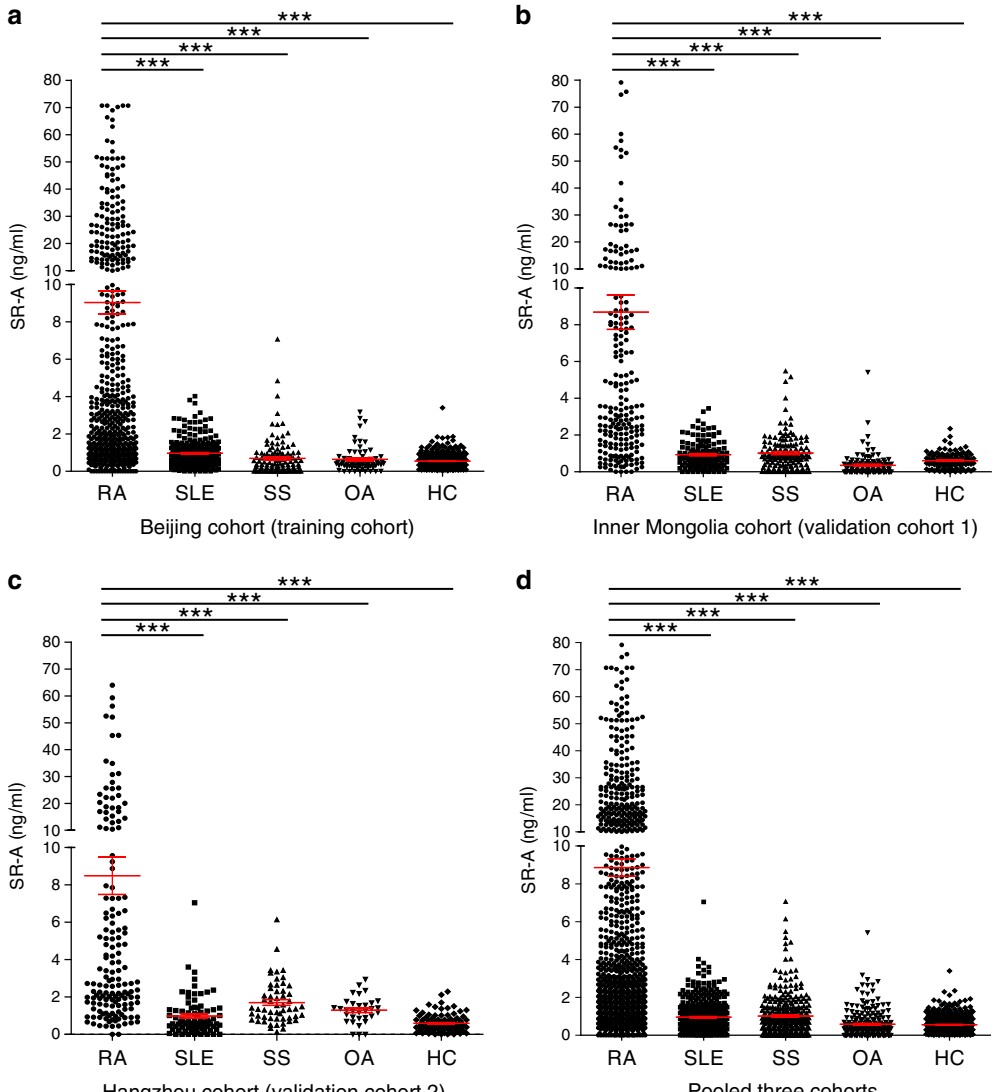

**Fig. 2 Detection of sSR-A in a large-scale, multicenter study.** A large-scale, multicenter study was performed to assess the discriminative power of serum sSR-A for RA using quantitative ELISA. The results of the training cohort, validation cohort 1, validation cohort 2, and the pooled three cohorts were shown, respectively. **a** Beijing cohort: training cohort (RA = 528; SLE = 254; SS = 120; OA = 72; HC = 480; ***all $p < 1E-15$), **b** Inner Mongolia cohort: validation cohort 1 (RA = 213; SLE = 144; SS = 144; OA = 119; HC = 120; ***all $p < 1E-15$), **c** Hangzhou cohort: validation cohort 2 (RA = 155; SLE = 80; SS = 55; OA = 32; HC = 100. ***$p < 1E-15$, $=2.1799E-04$, $=5.9038E-05$, and $<1E-15$, respectively, from left to right), **d** Pooled three cohorts: pooled all three cohorts' data together (***all $p < 1E-15$). Red horizontal lines: means; error bars: SEMs. ***$p < 0.001$ (Kruskal–Wallis test followed by Dunn's posttest for multiple comparisons). Source data are provided as a Source Data file.

those in ERA and RA patients, the levels of sSR-A were moderately increased in UA patients as compared with healthy controls (Fig. 5c). As shown in Fig. 5d, the prevalence of sSR-A in UA patients was 15.97% (19/119), comparable with anti-CCP (14.41%, 16/111) and RF (9.82%, 11/112). Moreover, those UA patients with high levels of sSR-A tended to display increased ESR or CRP, and positive RF. Comparing UA, ERA, and RA patients showed that the levels and/or positive rates of sSR-A were increased during disease progression (Fig. 5c). These findings indicate the potential value of sSR-A as a predictor of early RA. Yet follow-up studies are still needed to further validate the predictive value, which will be performed in our future work.

Taken together, all these results suggest that sSR-A represents a biomarker with the potential for an earlier and more accurate diagnosis of RA.

**Correlation of sSR-A with RA patient manifestations**. Correlation analysis showed that sSR-A was associated with many clinical and immunological features of RA patients (Table 2). Especially, the levels of sSR-A positively correlated with serum RF and IgM in both the training and validation cohorts. In Beijing cohort, the levels of sSR-A also correlated with glucose-6-phosphate isomerase (GPI). No obvious association was found between sSR-A and RA patient ages, CRP, and IgG.

We then divided the RA patients into sSR-A-positive and sSR-A-negative groups by the cut-off value. Detailed analyses showed that the levels of RF, IgM, and GPI were significantly higher in the sSR-A-positive group than in the sSR-A-negative group, consistent with the associations as described above (Supplementary Table 1).

There was also a modest correlation between sSR-A levels and RA patient radiographic damage as assessed by the Sharp/van der

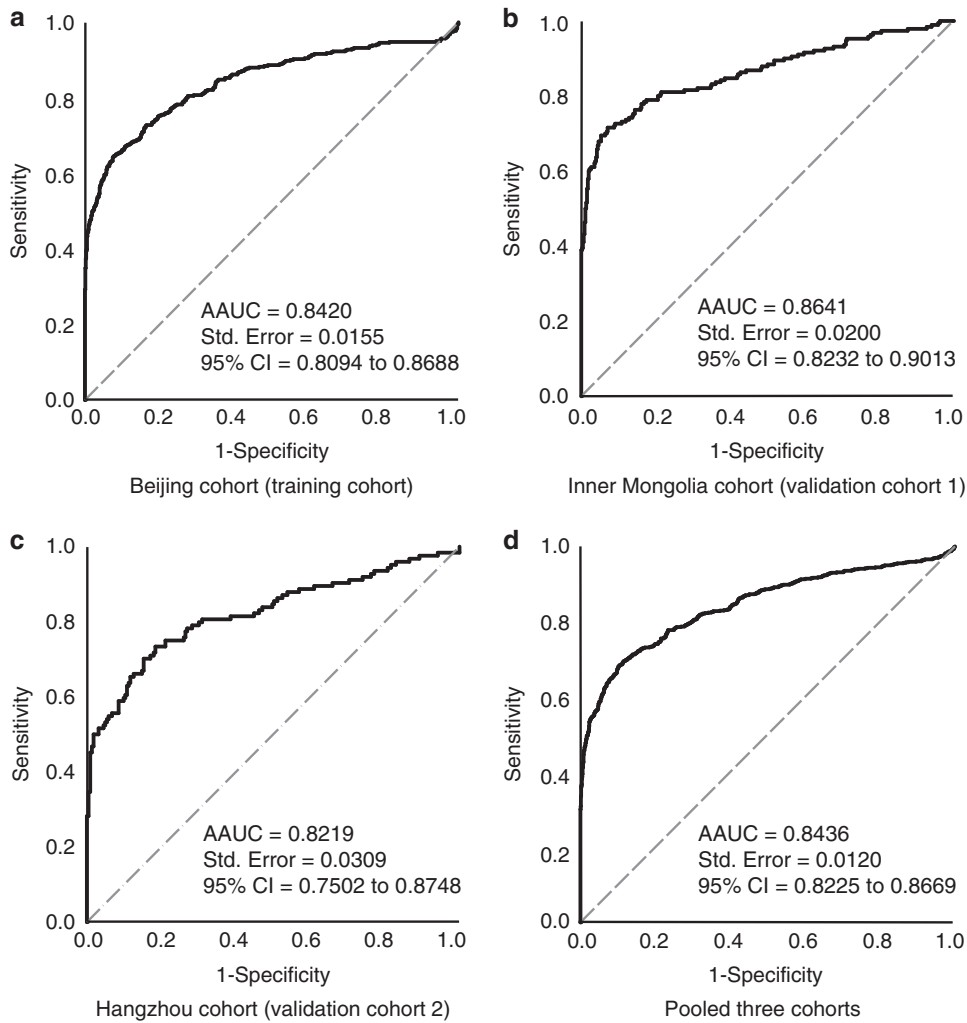

**Fig. 3 AROC curves of sSR-A for RA diagnosis.** Covariate-adjusted receiver operating characteristic curve (AROC) analysis was performed to evaluate the performance of sSR-A in diagnosis of RA, with age and gender as the confounders. The area under the age-adjusted and gender-adjusted ROC curve (AAUC) was 0.8420 (95% CI extending from 0.8094 to 0.8688) in Beijing cohort **a**, 0.8641 (95% CI extending from 0.8232 to 0.9013) in Inner Mongolia cohort **b**, 0.8219 (95% CI extending from 0.7502 to 0.8748) in Hangzhou cohort **c**, and 0.8436 (95% CI extending from 0.8225 to 0.8669) in the Pooled three cohorts **d**. AROC: covariate-adjusted receiver operating characteristic curve; AAUC: area under the covariate-adjusted ROC curve; CI: confidence interval (covariate-adjusted ROC curve analysis).

Heijde score (SHS, Supplementary Fig. 1a). Moreover, sSR-A-positive RA patients showed relatively higher SHS than sSR-A-negative RA patients (Supplementary Fig. 1b).

To further confirm these findings, we assessed the levels of sSR-A in both non-responders (DAS28 > 5.1) and responders (DAS28 < 2.6) of RA patients after therapy, and analyzed their clinical correlations, respectively. As shown in Supplementary Fig. 2, the levels of sSR-A were significantly decreased in the responders but not in the non-responders of RA patients after therapy. Moreover, these correlations as described above were more evident in the non-responders, yet could not be seen in the responders (Supplementary Table 2).

**Elevation of SR-A exacerbates autoimmune arthritis in mice.** We next investigated the role of sSR-A in disease pathogenesis using mouse arthritis models. Upon collagen-induced arthritis (CIA) in DBA/1 mice, there was a significant elevation of sSR-A in the serum as compared with that in naïve mice or adjuvant immunized mice (Fig. 6a). To further examine the activity of sSR-A, we i.v. injected recombinant SR-A protein (i.e., extracellular

domain of SR-A) to DBA/1 mice (2 μg/mouse) every 2 days starting from 2 days before boosting immunization for a total of five doses (Fig. 6b). Surprisingly, the mice receiving recombinant SR-A protein showed earlier disease onset as well as significantly higher clinical scores as compared with those control mice receiving saline (Fig. 6c, d). This disease-exacerbating effect was abolished when the recombinant SR-A protein was boiled prior to administration (Supplementary Fig. 3), excluding the possibility of endotoxin contamination. Supplementing SR-A protein in CIA mice also resulted in more severe bone destruction as assessed by micro-CT (Fig. 6e), and promoted the inflammation and pannus infiltrates in the joints (Fig. 6f). Evaluation of immune cells in lymphoid organs showed that injection of SR-A protein caused elevation of IL-17A in the serum associated with increased frequency of IL-17A-producing CD4[+] T cells (Fig. 6g, h), implicating a linkage of the sSR-A activity with a pathogenic Th17 response in arthritis. The activity of sSR-A in accelerating arthritis progression and Th17 inflammatory responses was also independently confirmed using a higher dose of SR-A protein (30 μg/mouse) (Supplementary Fig. 4).

**Table 1 Sensitivity and specificity of sSR-A in RA diagnosis.**

| Cohorts | Groups | N | Positive (n) | Sensitivity (%) | Specificity (%) | PPV (%) | NPV (%) |
|---|---|---|---|---|---|---|---|
| Beijing cohort (training cohort) | RA | 528 | 324 | 61.36 | 94.38 | 86.17 | 81.08 |
| | SLE | 254 | 31 | 12.20 | – | – | – |
| | SS | 120 | 11 | 9.17 | – | – | – |
| | OA | 72 | 5 | 6.94 | – | – | – |
| | HC | 480 | 5 | 1.04 | – | – | – |
| Inner Mongolia cohort (validation cohort 1) | RA | 213 | 156 | 73.24 | 90.51 | 75.73 | 89.33 |
| | SLE | 144 | 19 | 13.19 | – | – | – |
| | SS | 144 | 26 | 18.06 | – | – | – |
| | OA | 119 | 3 | 2.52 | – | – | – |
| | HC | 120 | 2 | 1.67 | – | – | – |
| Hangzhou cohort (validation cohort 2) | RA | 155 | 115 | 74.19 | 83.15 | 71.88 | 84.73 |
| | SLE | 80 | 13 | 16.25 | – | – | – |
| | SS | 55 | 23 | 41.82 | – | – | – |
| | OA | 32 | 7 | 21.88 | – | – | – |
| | HC | 100 | 2 | 2.00 | – | – | – |
| Pooled three cohorts | RA | 896 | 595 | 66.41 | 91.45 | 80.19 | 83.94 |
| | SLE | 478 | 63 | 13.18 | – | – | – |
| | SS | 319 | 60 | 18.81 | – | – | – |
| | OA | 223 | 15 | 6.73 | – | – | – |
| | HC | 700 | 9 | 1.29 | – | – | – |

The cut-off value of sSR-A was established 3 SD above the mean value of healthy controls. The sensitivity and specificity of sSR-A for identification of RA were 61.36% and 94.38% in Beijing cohort, 73.24% and 90.51% in Inner Mongolia cohort, 74.19% and 83.15% in Hangzhou cohort, and 66.41% and 91.45% in the pooled three cohorts.
RA rheumatoid arthritis, SLE systemic lupus erythematosus, SS Sjögren's syndrome, OA osteoarthritis, HC healthy control, N the number of total patients, n the number of sSR-A positive patients, PPV positive predictive value, NPV negative predictive value.

**SR-A ablation abrogates autoimmune arthritis in mice.** Using SR-A-deficient mice on a C57BL/6 background, we performed additional studies to validate the role of SR-A in arthritis development. Consistent with the results from DBA/1 mice, there was an evident increase of serum sSR-A during disease progression, which positively correlated with clinical scores of arthritis (Fig. 7a). Although our previous studies indicated an immunosuppressive function of SR-A in cancers[18,20,21] and hepatitis[23], we made a striking finding that ablation of SR-A rendered mice fully protected from CIA (Fig. 7b). Absence of SR-A also abolished cartilage erosion and inflammatory exudation in the articular cavity (Fig. 7c), which was associated with decreased infiltration of inflammatory myeloid cells and Th17 cells in the joint (Fig. 7d), further implicating SR-A as a factor that can drive inflammatory and pathogenic processes in arthritis. Additionally, absence of SR-A impaired the Th17 response in arthritic mice, indicated by reduced IL-17A levels in the blood (Fig. 7e), diminished production of IL-17A by splenocytes upon collagen stimulation (Fig. 7f), and decreased frequency of collagen-reactive Th17 cells in the spleens (Fig. 7g). Intracellular cytokine staining of splenocytes showed that, in addition to Th17 cells, lack of SR-A also dramatically reduced the frequency of Th1 cells, i.e., TNF-α-producing CD4$^+$ T cells (Fig. 7g), underscoring an important role of SR-A in arthritis-associated inflammation.

To exclude the possibility that SR-A deficiency might affect the intrinsic immunogenicity of antigen-presenting cells during collagen immunization, we examined DC activity and collagen-specific immune responses. No difference was seen between WT and SR-A$^{-/-}$ mice in the frequency and phenotype of DCs, collagen-reactive IFN-γ production (Supplementary Fig. 5), or a collagen-specific humoral response (Supplementary Fig. 6), suggesting that lack of SR-A in the model of CIA does not alter antigen-presenting function or antigen trafficking as previously reported[29].

To further investigate the relationship between IL-17A and sSR-A, we performed IL-17A neutralization in WT CIA mice. SR-A$^{-/-}$ CIA mice were also treated with IL-17A-neutralizing antibody as controls. It was shown that IL-17A neutralization substantially ameliorated the disease severity in WT mice (Supplementary Fig. 7a). However, neutralization of IL-17A failed to reduce the levels of serum sSR-A (Supplementary Fig. 7b). Additionally, incubation of bone marrow-derived macrophages (BMMφ) and bone marrow-derived dendritic cells (BMDCs), both of which highly express SR-A on cell surface, with inflammatory cytokines (i.e., IL-1β, IL-6, IL-17A, and TNF-α) failed to induce detectable secretion of sSR-A into the culture medium (Supplementary Fig. 8), suggesting that elevation of sSR-A may not be directly triggered by inflammatory cytokines.

**Inhibition of SR-A ameliorates the severity of arthritis.** Given the role of SR-A in promoting the disease progression and/or severity, we tested the hypothesis that SR-A can serve as a therapeutic target for RA. We showed that blockade of SR-A in CIA mice with either SR-A-neutralizing antibody or SR-A inhibitor Fucoidan (Fig. 8a) significantly reduced the severity of CIA (Fig. 8b). This SR-A inhibition also compromised the joint inflammation and bone destruction, as shown by the H&E staining and micro-CT imaging (Fig. 8c, d). Remission of clinical symptoms by SR-A inhibition was also associated with significantly decreased IL-17A in the serum (Fig. 8e). Function and pathology evaluation of the heart, liver, spleen, and kidney showed that no significant side effects or tissue damage were observed after treatment with SR-A-neutralizing antibody or Fucoidan (Fig. 8f, g).

**Discussion**

Scavenger receptor SR-A is an innate pattern recognition receptor with pleiotropic biological functions[13]. Previous studies have demonstrated important roles of SR-A in atherosclerosis[15,16], host response to pathogens[15,30], as well as antitumor immunity[18,20–22]. Our large-scale, multicenter study, presented here, has identified a soluble form of SR-A in the circulation as a potential RA diagnostic marker, which can also be used for complementary

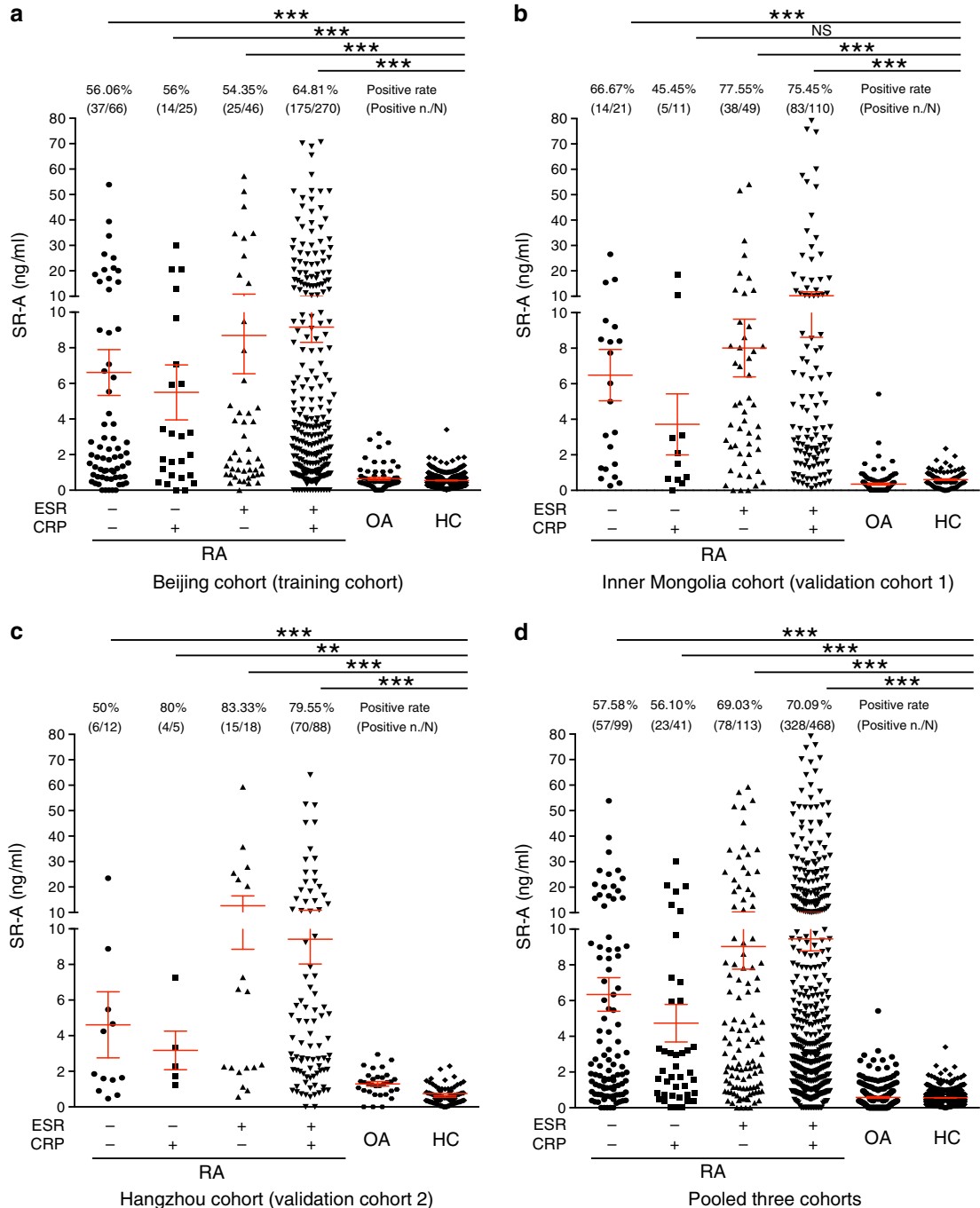

**Fig. 4 sSR-A in RA patients with normal ESR and/or CRP.** RA patients from the training and validation cohorts as well as the pooled cohort were divided into the following four groups, and the levels of sSR-A as well as the positive rates were further analyzed: RA patients with normal ESR (−) and normal CRP (−), RA patients with normal ESR (−) and increased CRP (+), RA patients with increased ESR (+) and normal CRP (−), RA patients with increased ESR (+) and increased CRP (+). **a** Beijing cohort: training cohort (***$p$ < 1E−15, =3.5748E−07, <1E−15, and <1E−15, respectively, from left to right), **b** Inner Mongolia cohort: validation cohort 1 (***$p$ = 4.6610E−06, = 6.7480E−11, and <1E−15, respectively, from left to right), **c** Hangzhou cohort: validation cohort 2 (**$p$ = 0.0069, ***$p$ = 1.6946E−04, = 1.2122E−10, and <1E−15, respectively, from left to right), **d** Pooled three cohorts (***$p$ < 1E−15, =5.6245E−10, <1E−15, and <1E−15, respectively, from left to right). $n$: the number of sSR-A-positive patients; $N$: the number of total patients. −: normal; +: increased. Red horizontal lines: means; error bars: SEMs. **$p$ < 0.01, ***$p$ < 0.001, NS, not significant (Kruskal–Wallis test followed by Dunn's posttest for multiple comparisons). Source data are provided as a Source Data file.

diagnosis of early RA as well as seronegative RA. More importantly, it also demonstrates potential as a predictor in UA patients. To the best of our knowledge, this is the first large-scale, multi-center study that reports the clinically diagnostic relevance of SR-A for RA in a training cohort and two independent validation cohorts.

The collective clinical data from three patient cohorts in different geographic regions shows that the discriminatory ability of sSR-A is comparable to the established biomarkers RF and anti-CCP, with a sensitivity and specificity of 61.36% and 94.38% in Beijing cohort, 73.24% and 90.51% in Inner Mongolia cohort, and 74.19% and 83.15% in Hangzhou cohort, respectively. The lower

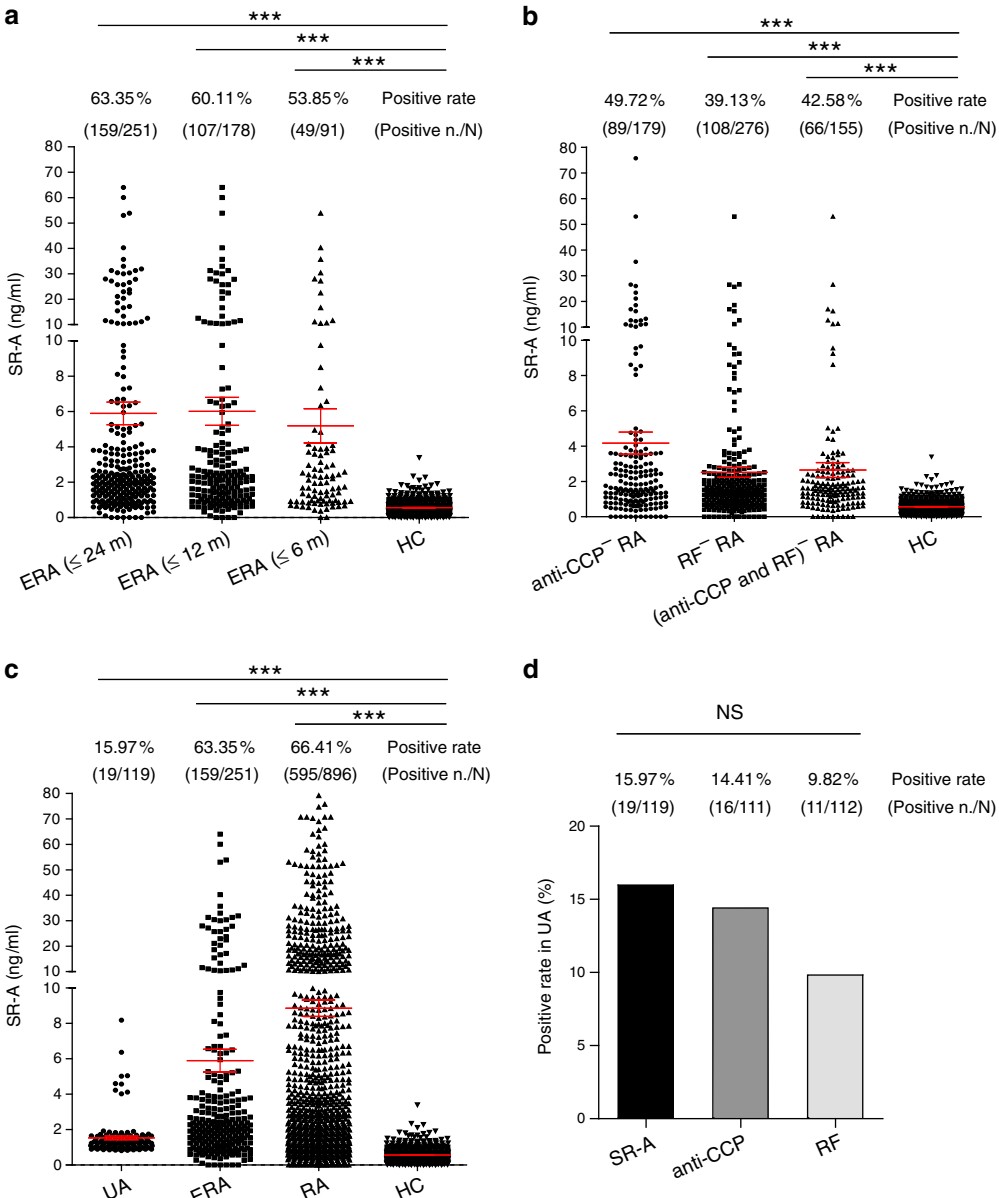

**Fig. 5 sSR-A in ERA, anti-CCP and/or RF-negative RA and UA patients. a** The levels of sSR-A in early RA (ERA) patients, including patients with disease duration <24 months ($n = 251$, ***$p < 1E−15$), <12 months ($n = 178$, ***$p < 1E−15$), and <6 months ($n = 91$, ***$p < 1E−15$) were detected. The positive rates of sSR-A were also analyzed. **b** The levels of sSR-A as well as the positive rates in anti-CCP and/or RF-negative RA patients, including anti-CCP⁻ RA patients ($n = 179$, ***$p < 1E−15$), RF⁻ RA patients ($n = 276$, ***$p < 1E−15$), and (anti-CCP and RF)⁻ RA patients ($n = 155$, ***$p < 1E−15$) were examined. **c** Comparing sSR-A levels and positive rates in UA ($n = 119$, ***$p < 1E−15$), ERA ($n = 251$, ***$p < 1E−15$) and RA ($n = 896$, ***$p < 1E−15$) patients. **d** Prevalence of sSR-A, anti-CCP, and RF in undifferentiated arthritis (UA) patients. $n$: the number of sSR-A-positive patients; $N$: the number of total patients. Red horizontal lines: means; error bars: SEMs. ***$p < 0.001$, NS, not significant (Kruskal–Wallis test followed by Dunn's posttest for multiple comparisons **a–c** or Chi-square test **d**). Source data are provided as a Source Data file.

specificity seen in Hangzhou cohort compares to that in Beijing and Inner Mongolia cohorts may be due to a small patient population in non-RA control group. Correlation analysis supports significant associations of sSR-A with IgM and RF in RA patients. Furthermore, our data support the complementary value of sSR-A in the diagnosis of anti-CCP and/or RF-negative RA. Therefore, combining sSR-A with currently established biomarkers may help improve the accuracy and sensitivity of RA diagnosis in the clinic.

Since early treatment, especially within 12 weeks of RA symptom onset, significantly improves the disease outcome, we have evaluated the diagnostic value of sSR-A in ERA. Our results demonstrate that even in RA patients with disease duration <6 months, sSR-A still shows a 53.85% positive rate, revealing high prevalence. Of note, there is also a 15.97% positive rate for sSR-A in UA patients, with a similar or higher prevalence than anti-CCP and RF. Integration of sSR-A into conventional RA diagnosis may potentially improve the early discrimination and/or the prediction of the disease, thereby improving the prognosis while reducing the co-morbidity and mortality.

It should be noted that most of findings in this study were derived from cross-sectional data, not from longitudinal data.

**Table 2 Association between sSR-A and RA patient clinical/immunological features.**

| Characteristics | Beijing cohort | | Inner Mongolia cohort | | Hangzhou cohort | |
|---|---|---|---|---|---|---|
| | r | p | r | p | r | p |
| Ages | 0.011 | 0.823 | −0.028 | 0.700 | 0.132 | 0.145 |
| Disease duration | 0.112 | 0.019* | 0.078 | 0.285 | 0.054 | 0.555 |
| ESR | 0.170 | 4.590E−04*** | 0.138 | 0.058 | 0.240 | 0.007** |
| CRP | 0.099 | 0.043* | 0.026 | 0.724 | 0.147 | 0.104 |
| DAS28 | 0.117 | 0.017* | 0.091 | 0.221 | 0.137 | 0.130 |
| IgA | 0.117 | 0.017* | 0.188 | 0.010* | 0.190 | 0.036* |
| **IgM** | **0.311** | **8.327E−11*** | **0.221** | **0.002**| 0.158 | 0.083 |
| IgG | 0.020 | 0.689 | 0.094 | 0.201 | 0.143 | 0.116 |
| **RF** | **0.622** | **7.599E−46*** | **0.506** | **1.215E−13*** | **0.584** | **1.288E−12*** |
| Anti-CCP | 0.293 | 3.313E−09*** | 0.083 | 0.261 | 0.145 | 0.115 |
| Albumin | −0.165 | 7.330E−04*** | −0.282 | 0.090 | – | – |
| WBC | 0.102 | 0.035* | 0.018 | 0.912 | – | – |
| **GPI** | **0.505** | **1.093E−26*** | – | – | – | – |

The important items with significant association are indicated by bold. *$p < 0.05$, **$p < 0.01$, ***$p < 0.001$, with exact $p$ values shown in the table (two-tailed Spearman's rank correlation test).
*ESR* erythrocyte sedimentation rate, *CRP* C-reactive protein, *DAS28* disease activity score 28, *Ig* immunoglobulin, *RF* rheumatoid factor, *Anti-CCP* anti-cyclic citrullinated peptide antibody, *WBC* white blood cell, *GPI* glucose-6-phosphate isomerase, *AKA* antikeratin antibodies, *APF* antiperinuclear factor antibodies.

Future studies using consecutive patients with undiagnosed joint pain are necessary to further evaluate the diagnostic value and predictive potential of sSR-A.

In addition to its diagnostic value, our current study has revealed an unexpected role of sSR-A in promoting autoimmune pathogenesis using animal model of RA (i.e., CIA). Consistent with our observations in RA patients, the heightened level of sSR-A positively correlates with disease severity in both C57BL/6 and DBA/1 mice that have developed experimental arthritis. Furthermore, supplementing SR-A protein accelerates the arthritic development and progression concomitant with increased T cell activation (i.e., Th17 response) in mice. On the contrary, inhibition of SR-A by neutralizing antibody or inhibitor can reduce incidence and/or ameliorate the disease symptoms of CIA. We have validated the pathogenic activity of SR-A using genetically deficient mouse mode. $SR\text{-}A^{-/-}$ mice are resistant to the induction of CIA, evidenced by sharply reduced joint pathology and autoimmune inflammation. Increased resistance to CIA in the absence of $SR\text{-}A$ is associated with impaired IL-17A and TNF-α production by helper T cells, which are known to be major contributors to CIA[31,32].

SR-A is highly expressed on differentiated myeloid cells (e.g., macrophages) and immature myeloid cells that have been reported to promote pathogenesis of RA[33,34]. Given that genetic ablation abolishes both cell-associated SR-A (i.e., cSR-A) and sSR-A, the possibility that cSR-A is also involved in RA pathogenesis cannot be excluded. However, absence of cSR-A in DCs does not appear to affect their activation and subsequent induction of collagen-specific T cells upon vaccination in CIA model. Additionally, the molecular size of sSR-A in the circulation is smaller than cSR-A and various matrix metalloproteinases (e.g., MMP-3, MMP-9, and ADAM-17) are up-regulated in RA patients (unpublished data), suggesting that sSR-A may be derived from the cleavage of cSR-A. However, the molecular mechanisms underlying the production of sSR-A in RA remain to be determined. Intriguingly, although IL-17A neutralization substantially alleviates the disease severity of CIA, it does not reduce, but slightly increases, the levels of sSR-A in vivo, indicating that elevation of sSR-A may require additional signals and occur prior to amplified Th17 response and IL-17A production. Moreover, inflammatory cytokines (i.e., IL-1β, IL-6, IL-17A, and TNF-α) appear to have little effect on the production of sSR-A from myeloid cells. These suggest that inflammatory factors may

not directly trigger the cellular release of sSR-A despite that sSR-A represents a pathogenic factor capable of promoting disease progression and amplifying inflammatory response in RA. Our findings also partially explain why sSR-A is not increased in AS patients in which IL-17A plays an indispensable role. Nevertheless, the involvement of SR-A in AS pathogenesis needs to be further investigated.

Together, our findings from RA patients support the use of serum sSR-A for RA diagnosis, especially in anti-CCP and/or RF-negative RA patients. Moreover, use of sSR-A for RA early diagnosis and UA prediction could benefit the patients by allowing their earlier access to treatment, thereby preserving the joint functions and minimizing economic cost. Considering a potential role of SR-A in RA pathogenesis, indicated by our data from mouse models of RA, treatment strategies targeting human SR-A may be developed for management of RA.

## Methods

**Study population and serum samples.** Serum samples from 1117 RA patients (including 251 early RA, 179 anti-CCP-negative RA, 276 RF-negative RA, and 155 anti-CCP & RF-negative RA), 478 SLE patients, 319 SS patients, 223 OA patients, 119 UA patients, 39 AS patients, 39 Gout patients, 39 PsA patients, 39 AAV patients, 30 AOSD patients, 24 PMR patients, 25 AIH patients, 71 non-autoimmune inflammatory disease patients, and 700 healthy volunteers were collected between 2013 and 2018 at the Department of Rheumatology and Immunology, Peking University People's Hospital, Beijing, China, the Department of Rheumatology and Immunology, First Hospital Affiliated to Baotou Medical College, Inner Mongolia, China, and the Department of Rheumatology, the Second Affiliated Hospital, Zhejiang University, Zhejiang, China. Demographics and clinical characteristics of all patients from the three training and validation cohorts are provided in Supplementary Tables 3–8.

While RA patients met the 2010 American College of Rheumatology (ACR) and European League Against Rheumatism (EULAR) criteria for RA, all SLE (SLICC Revision of ACR 2009), SS (ACR 2012), OA (ACR 1995), AS (Modified New York criteria 1984), Gout (ACR/EULAR 2015), PsA (CASPAR 2006), AAV (CHCC 2012), AOSD (Yamaguchi 1992), PMR (EULAR/ACR 2015), UA (EULAR 2007), AIH (IAIHG 1999), and non-autoimmune inflammatory disease patients fulfilled their classification criteria, respectively. The study was approved by the Research Ethics Committee at Peking University People's Hospital, Beijing, China, the Ethical Review Board of the First Hospital Affiliated to Baotou Medical College, Inner Mongolia, China, and the Ethical Review Board of the Second Affiliated Hospital, Zhejiang University, Zhejiang, China. Informed consent was obtained from all patients and healthy donors.

**ELISA analysis of sSR-A.** Sera from patients or arthritic mice were aliquoted and stored at −80 °C until use. Commercially available human SR-A ELISA kit (Sino Biological Inc., Beijing, China) and mouse SR-A ELISA kit (USCN, Wuhan, China) were used according to the manufacturer's instructions with modifications. The

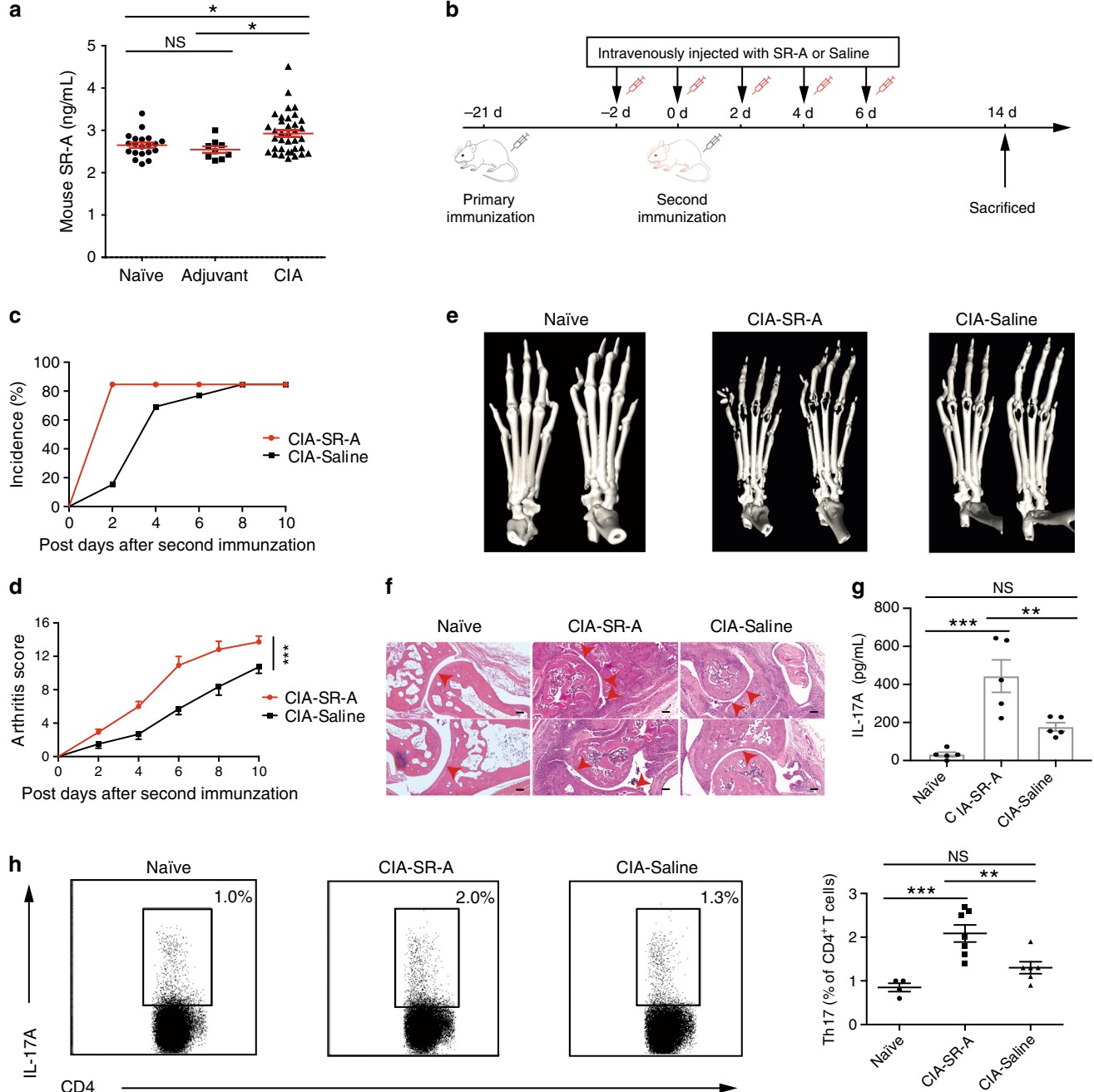

**Fig. 6 Elevation of sSR-A accelerates arthritis onset and exacerbates disease severity in mice. a** The serum levels of mouse sSR-A were examined in the collagen-induced arthritis (CIA) and control mice by ELISA. The sSR-A levels were significantly higher in CIA mice than in adjuvant immunized or untreated DBA/1 mice (Naïve, $n = 20$; Adjuvant, $n = 9$; CIA, $n = 36$. *$p = 0.0422$ (left) and 0.0343 (right)). **b** Scheme of the experimental setup. DBA/1 mice were intravenously injected with recombinant SR-A protein (5 μg/mouse) or Saline every 2 days, starting from 2 days before boosting immunization for a total of five times. Mice were monitored every 2 days and sacrificed 2 weeks after the second immunization. **c, d** The arthritis incidence and the clinical scores were shown for both CIA-Saline and CIA-SR-A mice over time (CIA-SR-A, $n = 13$; CIA-Saline, $n = 13$, ***$p = 2.4276E{-}04$). **e** Micro-CT image showing the bone destruction of paws from Naïve, CIA-SR-A, and CIA-Saline mice. **f** H&E staining of the sagittal sections of paws from Naïve, CIA-SR-A, and CIA-Saline mice. Arrows indicate the stenosis of articular cavity and destruction of cartilage. Scale bar = 100 μm. **g** The serum IL-17A levels were examined in Naïve, CIA-SR-A, and CIA-Saline mice ($n = 5$ per group, **$p = 0.0082$, ***$p = 2.8143E{-}04$). **h** The frequency of Th17 cells in the spleen was analyzed by flow cytometry. The representative flow charts and the statistical results were shown (Naïve, $n = 4$; CIA-SR-A, $n = 7$; CIA-Saline, $n = 6$, **$p = 0.0093$, ***$p = 6.5803E{-}04$). Data are presented as mean ± SEM. Results are representative of three independent experiments. *$p < 0.05$, **$p < 0.01$, ***$p < 0.001$, NS, not significant (one-way ANOVA test followed by Tukey's posttest for multiple comparisons **a**, **g**, **h** or two-way repeated measures ANOVA test **d**). Source data are provided as a Source Data file.

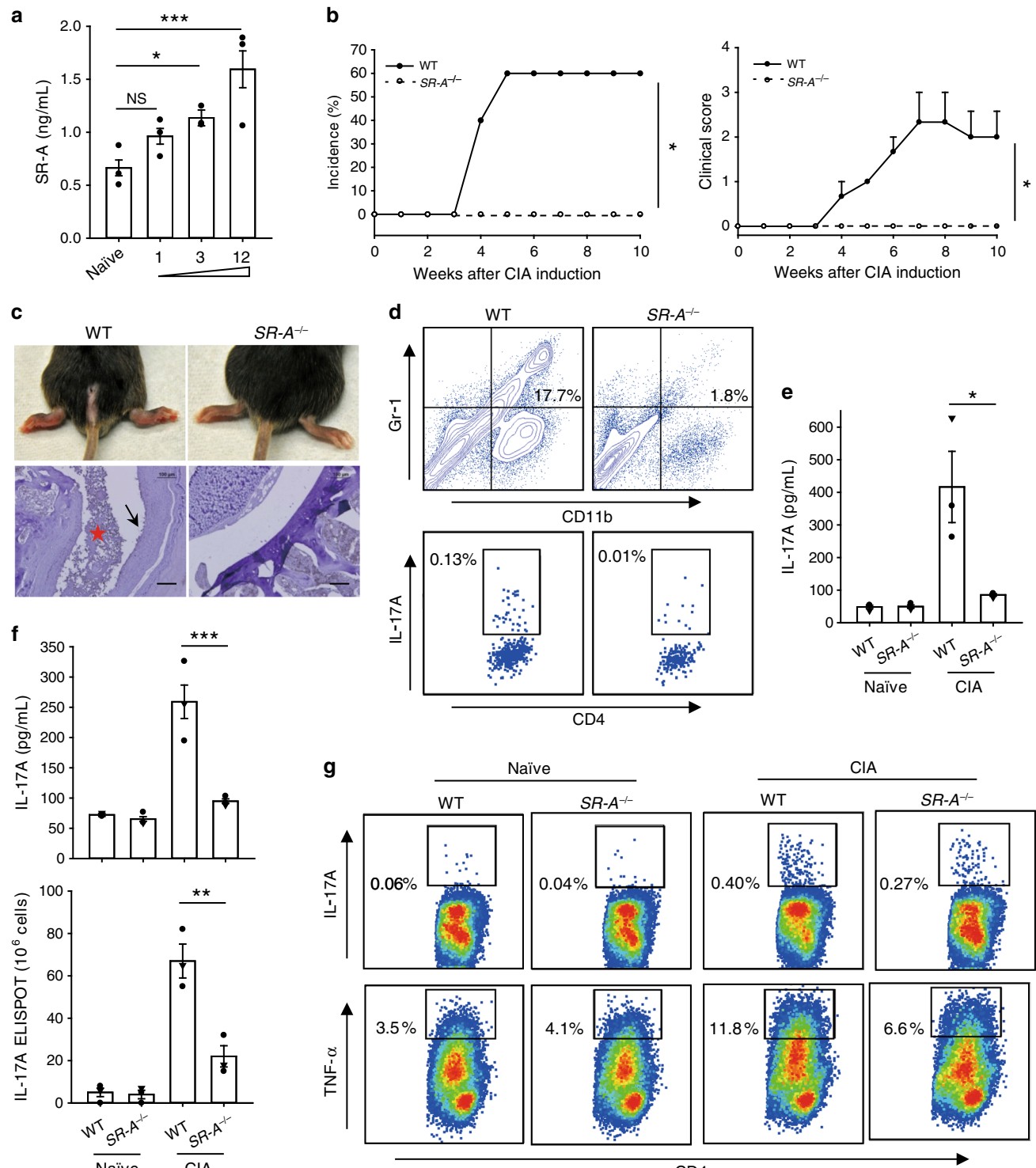

**Fig. 7 Reduced incidence of CIA and autoimmune inflammation in *SR-A*-deficient mice. a** The serum levels of mouse sSR-A in naïve C57BL/6 mice or CIA mice with different clinical scores were examined by ELISA ($n = 3$ per group, $*p = 0.0155$, $***p = 1.7785E-05$). **b** Male WT ($n = 5$) and $SR-A^{-/-}$ ($n = 5$) mice were immunized with 200 μg chicken collagen II emulsified in CFA. The arthritis incidence ($*p = 0.049$) and disease severity ($*p = 0.0171$) were followed. **c** Representative gross image of arthritis symptoms was recorded at 6 weeks post immunization. Inflammatory cell infiltration and bone erosion were evaluated by H&E staining. Red star indicates inflammatory cell infiltration. Arrow indicates bone erosion. Scale bar = 100 μm. **d** The infiltration of MDSCs and Th17 cells into the inflamed joints was determined by flow cytometry. **e** Six weeks after CIA induction, serum level of IL-17A in the arthritic mice ($n = 3$ per group, $*p = 0.0387$) was determined by ELISA. **f–g** Splenocytes from naïve or CIA WT/$SR-A^{-/-}$ mice were cultured in the presence of denatured collagen (50 μg/mL) for 48 h. IL-17A levels in the culture medium were determined by ELISA (**f**, top, $***p = 1.5376E-04$). Frequencies of IL-17A-producing cells were evaluated by ELISPOT (**f**, bottom, $**p = 0.0085$) or intracellular cytokine staining (**g**, top). The frequencies of TNF-α-producing CD4$^+$ T cells upon collagen stimulation were also assayed by intracellular cytokine staining (**g**, bottom). Data are presented as mean ± SEM. Results are representative of three independent experiments. $*p < 0.05$, $**p < 0.01$, $***p < 0.001$, NS, not significant (one-way ANOVA test followed by Dunnett's posttest for multiple comparisons **a**, LogRank test and two-way repeated measures ANOVA test **b**, or two-tailed Student's $t$ test **e** and **f**). Source data are provided as a Source Data file.

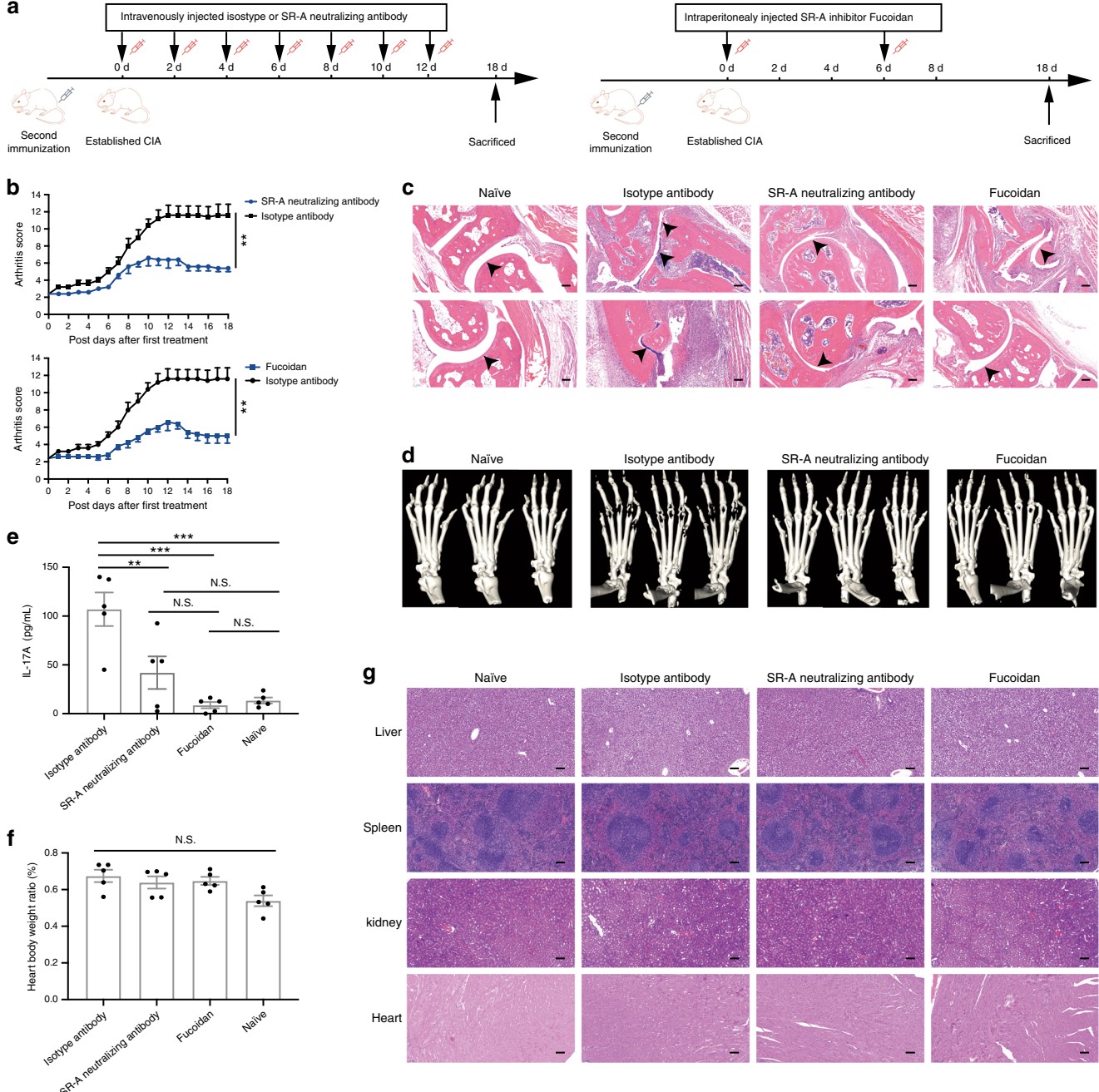

**Fig. 8 Inhibition of SR-A ameliorates severity of arthritis in mice. a** Scheme of the experimental setup. Intervention was initiated when the arthritis scores were averagely over 2. CIA mice were intravenously injected with SR-A neutralizing antibody or isotype IgG (2 μg/mouse) once every 2 days for seven doses, or were intraperitoneally injected with SR-A inhibitor Fucoidan (100 μg/mouse) on day 0 and day 6 after initiation for two doses. **b** The arthritis scores in each intervention group were recorded daily after the first treatment ($n = 5$ per group, **$p = 0.0028$ (top) and 0.0016 (bottom), respectively). **c** H&E staining of the sagittal sections of paws from the isotype antibody-, SR-A neutralizing antibody-, or Fucoidan-treated CIA mice and naïve mice. Arrows indicate the stenosis of articular cavity and destruction of cartilage. Scale bar = 100 μm. **d** Micro-CT image showing the bone destruction of paws from the isotype antibody-, SR-A neutralizing antibody-, or Fucoidan-treated CIA mice and naïve mice. **e** The serum IL-17A levels were detected in the isotype antibody-, SR-A neutralizing antibody-, or Fucoidan-treated CIA mice and naïve mice ($n = 5$ per group, **$p = 0.0081$, ***$p = 1.7201\text{E}{-}04$ (left) and 2.9233E−04 (right)). **f** The heart body weight ratio was shown to reflect the heart functions ($n = 5$ per group). **g** H&E staining of the sagittal sections of liver, spleen, kidney, and heart were performed to evaluate the organ damage. Scale bar = 100 μm. Data are presented as mean ± SEM. Results are representative of three independent experiments. **$p < 0.01$, ***$p < 0.001$, NS, not significant (two-way repeated measures ANOVA test **b** or One-way ANOVA test followed by Tukey's posttest for multiple comparisons **e**, **f**). Source data are provided as a Source Data file.

results were measured on a Synergy™ 4 Multi-Mode Microplate Reader with software GEN5CH 2.0 (BioTek, Winooski, VT).

**Diagnostic value analysis of sSR-A for RA.** Multivariable logistic regression analysis was first performed to identify the confounders between sSR-A and RA with the statistical software SPSS 25 (SPSS Inc., Chicago, IL). Then the covariate-adjusted receiver-operating characteristic curve (AROC) analysis was performed with the statistical software StataSE 15 (StataCorp, College Station, TX) to evaluate the diagnostic utility of sSR-A using non-parametric method, as estimated by the area under the covariate-adjusted ROC curve (AAUC). The corresponding Stata programs were developed by Dr. Pepe's group[25–27], and could be found at the following website: https://research.fhcrc.org/diagnostic-biomarkers-center/en/software.html. The optimal positive cut-off value in the study was set for 3 SD above the mean value of the healthy controls, which showed better clinical utility of sensitivity and specificity than the ROC curve and Youden index analysis.

**Mice.** Six to eight-week old male DBA/1 mice were obtained from Huafukang Bioscience Company (Beijing, China). Ten-week old male C57BL/6, $SR-A^{-/-}$, and DBA/1 mice were obtained from the Jackson Laboratory (Bar Harbor, ME). All mice were housed in specific pathogen-free environment under controlled conditions (22 °C ambient temperature, 40% humidity). All animal procedures complied with relevant ethical regulations for animal testing and research, and were approved by the institutional animal care and use committee (IACUC) of Peking University People's Hospital and Virginia Commonwealth University.

**Reagents.** Fluorochrome-conjugated monoclonal antibodies (mAbs), including PE-CD11b (M1/70, 101207, 1:400), PerCP/Cy5.5-IL-17A (TC11-18H10.1, 506919, 1:100), PE/Cy5-Gr-1 (RB6-8C5, 108409, 1:600), APC-CD4 (GK1.5, 100412, 1:400), FITC-TNF-α (MP6-XT22, 506304, 1:100), and APC/Fire750-CD4 (GK1.5, 100460, 1:100) were purchased from BioLegend (San Diego, CA), while PE-IL-17A (eBio17B7, 12-7177-81, 1:100) was purchased from eBioscience (San Diego, CA). Anti-mouse IL-17A neutralizing monoclonal antibody (17F3, BP0173) was purchased from BioXCell (West Lebanon, NH). Recombinant mouse IL-1β, IL-6, IL-17A, and TNF-α were purchased from BioLegend. Mouse IL-17A ELISA kit was purchased from BioLegend. Mouse IL-17A ELISPOT kit was purchased from R & D Systems (Minneapolis, MN). Native chicken and bovine collagen type II were purchased from Sigma-Aldrich (St. Louis, MO) and Chondrex (Redmond, WA), respectively.

**Experimental arthritis.** CIA models were established in DBA/1 mice by immunization on day 1 and 21 (200 and 100 μg bovine type II collagen, respectively). The severity of arthritis was scored based on the level of inflammation in each of the four paws and recorded as one of four grades: 0, normal; 1, erythema and swelling of one or several digits; 2, erythema and moderate swelling extending from the ankle to the mid-foot (tarsals); 3, erythema and severe swelling extending from the ankle to the metatarsal joints; and 4, complete erythema and swelling encompassing the ankle, foot, and digits, resulting in deformity and/or ankyloses. The scores of all four limbs were summed, yielding a total score of 0–16 per mouse.

Arthritis induction in C57BL/6 mice was also performed[33,35]. Briefly, mice were subjected to a single intradermal injection of 200 μg native chicken type II collagen (4 mg/mL in 0.05 M acetic acid) emulsified in an equal volume of Complete Freund's Adjuvant containing 4 mg/mL heat-killed *Mycobacterium tuberculosis* H37 (Difco laboratories, Detroit, MI) at the base of the tail. The severity of arthritis was scored for each limb[36].

**Injection of recombinant SR-A protein and SR-A inhibitors.** DBA/1 mice established with arthritis received intravenous injection of extracellular domain of mouse SR-A protein at dose of 5 or 30 μg/mouse every 2 days, starting from 2 days before boosting immunization for five doses. The control mice were treated with equal volume of physiological saline.

Injection of SR-A-neutralizing antibody or inhibitor was initiated when the arthritis scores were averagely over 2. CIA mice were intravenously injected with SR-A-neutralizing antibody (AF1797, R & D Systems; 2 μg/mouse in 200 μL PBS) or isotype IgG once every 2 days for seven doses, or were intraperitoneally injected with SR-A inhibitor fucoidan (Sigma; 100 μg/mouse in 200 μL PBS) on day 0 and day 6 after initiation for two doses.

**Mouse paw histopathology and micro-CT.** For histological analysis, mice were sacrificed and tissues were collected and fixed in 4% buffered formaldehyde. Tissues were then paraffin embedded, sectioned, stained with H&E, and analyzed with NDP.view2 (Hamamatsu Photonics K.K., Japan). Micro-CT images were acquired on the Tri-Modality FLEX Triumph™ Pre-Clinical Imaging System (Gamma Medica-Ideas, Northridge, CA). CT image sets acquisitions lasted 10 min and utilized beam parameters of 130 μA and 80 kVP. Analyze 10.0 (AnalyzeDirect, Overland Park, KS) was used to perform the image analysis.

**Flow cytometry.** For surface staining, cells were stained with various antibodies at room temperature for 30 min, washed with PBS and then re-suspended and fixed

with 1.5% formaldehyde for 2 min under room temperature. For intracellular staining, cells were incubated with PMA (50 ng/mL, Multisciences, China), ionomycin (1 μg/mL, Multisciences, China), and BFA (10 μg/mL, Multisciences, China) for 5 h, then surface stained, fixed, permeabilized, and intracellular stained in accordance with the manufacture's instruction. Cells were analyzed on FACS Arial II flow cytometer with FACSDiva Software V6.1.3 (Becton Dickinson, San Diego, CA). Results were further analyzed by FlowJo 10 data analysis software (FLOWJO, LLC Ashland, OR).

Gating strategies are presented in Supplementary Fig. 9.

**Statistical analysis.** Statistical calculations were performed using the statistical software programs GraphPad Prism 8 (GraphPad Software Inc., San Diego, CA), SPSS 25 (SPSS Inc.), SAS 8.1 (SAS Institute Inc., Cary, NC), StataSE 15 (StataCorp, College Station, TX) or SigmaPlot 12.5 (Systat Software Inc., San Jose, CA). Differences between various groups were evaluated by the Student's $t$ test, One-way ANOVA test, Mann–Whitney $U$ test, Kruskal–Wallis H test, Chi-square test, Spearman's rank correlation test, LogRank test or two-way repeated measures ANOVA test. All statistical analyses with $p$ value < 0.05 were considered statistically significant (*$p$ < 0.05, **$p$ < 0.01, ***$p$ < 0.001, N.S., not significant).

**Reporting summary.** Further information on research design is available in the Nature Research Reporting Summary linked to this article.

## Data availability

The authors declare that all data supporting the findings of this study are available within the article and its Supplementary Information files or are available from the authors on request. The source data underlying Figs. 1a–d, 2a–d, 4a–d, 5a–c, 6a, c, d, g, h, 7a, b, e, f, 8b, e, f, and Supplementary Figs. 1b, 2, 3, 4a, b, 5c, 6, 7a, b, 8 are provided as a Source Data file.

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

## Acknowledgements

We thank W. Guo and L. Liu for assistance of µCT analysis, as well as H. Liu for assistance of statistical analysis. This work was supported by grants from the National Natural Science Foundation of China (81971523 and 81671604 to FL.H., 31530020 to Z.L., and 81871281 to Y.J.), the Beijing Nova Program (Z181100006218044 to FL.H.), and the Beijing Municipal Science & Technology Commission (Z171100000417007 to Z.L. and Z191100006619109 to Y.S.), as well as by Macau Science and Technology Development Fund (0094/2018/A3), Peking University Clinical Medicine Plus X-Young Scholars Project Funds (PKU2019LCXQ018 to FL.H.), Virginia Commonwealth University Research Development Funds and US Department of Defense Award (W81XWH1910538 to X.-Y.W.). Flow cytometry facility at VCU was supported in part by NCI Cancer Center Support Grant to VCU Massey Cancer Center P30CA16059. The funders had no role in study design, data collection and analysis, decision to publish, or preparation of the manuscript.

## Author contributions

Performed the experiments: FL.H., X.J., and C.G.; Analyzed the data: FL.H., X.J., C.G., S.C., P.W., X.Z., and Y.X.; Contributed reagents/materials/analysis tools: Y.L., W.Z., Y.D., X.F., X.L., J.S., F.H., J.X., M.B., Y.J., X.Liu, L.R., XY.Z., J.G., H.P., Y.S., H.Yi, H.Y., D.Z., J.L., H.W., Y.W., R.L., and L.L.; Wrote the manuscript: FL.H., X.J., and C.G.; Reviewed and edited the manuscript: X-Y.W. and Z.G.L.

## Competing interests

The authors declare no competing interests.
