## [Peer Review File · Nature Communications]

Reviewers' comments:

Reviewer #1 (Remarks to the Author):

This manuscript aims to test scavenger receptor-A (SR-A), also termed CD204, as a diagnostic biomarker of RA. It is a very well written manuscript and a very interesting read for those interested in the pathophysiology of RA. There is a clinical unmet need for such a biomarker to enhance early diagnosis and thereby institute early intervention with disease-modifying therapies. The following evidence is presented in support of this biomarker:

1. Levels of SR-A are increased compared to healthy individuals and disease controls that may mimic RA. The performance in early RA cohorts is similar to anti-CCP and additional recently described biomarkers such as 14-3-3eta. A major limitation of this assessment is the lack of data in RA patients who lack anti-CCP or RF antibodies. In Table 2, only about 20% of anti-CCP and RF negative patients are SR-A positive amongst 39 patients tested in Beijing. This could add diagnostic value but it is difficult to place confidence in this data because the sample size is too small. It is unclear how this biomarker enhances PPV/NPV when added to RA antibodies. What is also lacking is data from consecutive patients presenting with undiagnosed arthralgia suspicious for RA.
2. The levels of SR-A decline with treatment suggesting the possibility that the biomarker could be a disease activity biomarker. However, it is not indicated to what degree the biomarker enhances the performance of currently used outcomes for measuring disease activity e.g. CRP.
3. Mice that receive the recombinant protein experience earlier disease onset and enhanced disease severity and more destruction of bone as measured by micro-CT. This was associated with elevated levels of IL17A although abrogation of the effect of SR-A with anti-IL17A was not tested. SR-A deficient mice on a C57BL/6 background develop less severe disease. Lack of SR-A also reduced the levels of TNF- α -producing CD4+ T cells.
4. The severity of CIA was reduced with SR-A antibody mediated blockade.

Overall, the manuscript provides compelling evidence for a role of this mediator in the development of CIA but provides less compelling data for the value of this biomarker for diagnosis. The sample size for early RA and anti-CCP/RF negative RA is quite low. The PPV falls below 80% which would be considered a reasonable cut-off for assigning significant clinical utility. The data primarily emanates from well-established disease cohorts and the value of the test in patients presenting with arthralgia is unclear. It would be interesting to examine the value of this biomarker as a prognostic indicator. It would have helped the manuscript enormously if the authors had provided data from a longitudinal cohort attesting to its value as a prognostic indicator, especially for radiographic damage. That is where I think there is the most important unmet clinical need in the field, a modifiable prognostic biomarker. The data in CIA presented in this work suggests there could be a key role for SR-A as a prognostic biomarker.

Reviewer #2 (Remarks to the Author):

Hu F., et al measured the serum soluble scavenger receptor A in a large-scale test and validation cohorts and assessed the usefulness as a potential biomarkers to support a diagnosis of rheumatoid arthritis. In addition, they tested the role of scavenger receptor A in collagen-induced arthritis model. Although the topic is interesting, there are a list of concerns below.

1. What is the rationale to identify this biomarker as a diagnostic biomarker? How did the authors find this biomarker as one of the best biomarkers among a long list of potential biomarkers? How did the authors collaborate that this biomarker is superior to many other biomarkers reported so far?

2. The authors should provide sufficient information about the treatment on RA and clinical characteristics of disease controls.
3. In ACR/EULAR classification criteria, the rationale and golden standard to generate criteria is, first, to identify clinical and laboratory variables to predict initiation of DMARDs in undifferentiated arthritis, and then the potential variables are tested for the relative contribution in influencing the probability of developing 'persistent synovitis and/or erosive arthritis' as developing RA by using real-life case scenarios. Given these work flows, are the patients cohort used in this study appropriate to identify the new biomarkers to help early diagnosis of RA? In order to confirm the performance of biomarkers to early diagnosis, the authors should test the potential biomarkers in undifferentiated arthritis cohort and show the comparable or superior performance to the pre-existing laboratory variables.
4. In ACR/EULAR2010 classification criteria, one has to exclude other diseases that best explained the signs and symptoms. In this regard, did the authors include appropriate disease controls to test specificity of the potential biomarkers including diseases with higher levels of inflammation such as polymyalgia rheumatic, adult onset still's disease, vasculitis and so on?
5. In ACR/EULAR 2010 classification criteria, CRP and ESR are listed in addition to anti-CCP and RF. Is the soluble scavenger receptor A superior to CRP and ESR? How did weigh this potential biomarker in the classification criteria?
6. Is the correlation between soluble SR-A and other clinical variables true for the inception cohort? The authors should demonstrate the correlation not only by cross-sectional cohorts, but also by longitudinal cohorts consisting of appropriate and sufficient numbers of the patients with responders and non-responders.
7. What is the biological role of soluble scavenger receptor on the pathogenesis of rheumatoid arthritis? Is this molecule mediating the persistent inflammation in synovium and involving in the joint destruction? Is this molecule regulated by inflammatory cytokines such as TNF-alpha and/ or IL-6?
8. In collagen-induced arthritis model without anti-citrullinated protein antibodies, are the levels of soluble scavenger receptor A correlating with the joint destruction, in addition to joint counts? Is the inhibition of SR-A ameliorating joint destruction? By blocking with anti-IL17A monoclonal antibodies, are the arthritis and joint destruction comparable to those inhibited by SR-A knock out? Are levels of soluble SR-A down-regulated by IL-17A? Given the apparent difference between human RA and CIA for the role of IL-17A, along with the indispensable role of IL-17 A in ankylosing spondylitis, the authors should comment on the role of SR-A in human RA and AS.
9. There are so many inconsistencies for the relationship between soluble SR-A and RF and the positive and negative role of soluble form of SR-A on T cell function. The authors should clarify the functional role of soluble form and membrane bound form of SR-A.

Detailed responses to the reviewers' comments:

We appreciate the valuable suggestions of the expert reviewers. We have conducted additional experiments and included new data in the revised manuscript, which we believe have addressed the reviewers' concerns and improved our manuscript significantly. The point-to-point responses to each question are as follows:

Reviewer #1:

This manuscript aims to test scavenger receptor-A (SR-A), also termed CD204, as a diagnostic biomarker of RA. It is a very well written manuscript and a very interesting read for those interested in the pathophysiology of RA. There is a clinical unmet need for such a biomarker to enhance early diagnosis and thereby institute early intervention with disease-modifying therapies. The following evidence is presented in support of this biomarker:

1. Levels of SR-A are increased compared to healthy individuals and disease controls that may mimic RA. The performance in early RA cohorts is similar to anti-CCP and additional recently described biomarkers such as 14-3-3eta. A major limitation of this assessment is the lack of data in RA patients who lack anti-CCP or RF antibodies. In Table 2, only about 20% of anti-CCP and RF negative patients are SR-A positive amongst 39 patients tested in Beijing. This could add diagnostic value but it is difficult to place confidence in this data because the sample size is too small. It is unclear how this biomarker enhances PPV/NPV when added to RA antibodies. What is also lacking is data from consecutive patients presenting with undiagnosed arthralgia suspicious for RA.

Response:

To further demonstrate the diagnostic value of soluble SR-A (sSR-A) in RA, we have conducted additional studies and included the following results to the revised manuscript.

- (1) Diagnostic value of sSR-A in anti-CCP and/or RF negative RA: As suggested, we enlarged the sample size of anti-CCP and/or RF negative RA patients. Serum samples from 179 anti-CCP-negative RA patients, 276 RF-negative RA patients, and 155 (anti-CCP+RF)-double negative RA patients were subjected to the detection of sSR-A levels and prevalence. The positive rates of sSR-A in anti-CCP-negative and RF-negative RA patients were 49.72% (89/179) and 39.13% (108/276), respectively. More importantly, sSR-A also showed a 42.58% (66/155) prevalence in (anti-CCP+RF)-double negative RA patients. These results support the use of sSR-A to complement the diagnosis in anti-CCP and/or RF negative RA patients. The results are described on Page 10 of the revised manuscript and shown in Figure 5B.
- (2) Diagnostic value of sSR-A in early RA (ERA): We further enlarged the sample size of ERA patients in our study, including 251 ERA with disease duration less than 24 months, 178 ERA with disease duration less than 12 months, and 91 ERA with disease duration less than 6 months. We showed that in these ERA patients, sSR-A showed powerful diagnostic value. In early RA patients with disease duration less than 12 and 24 months,

the positive rates of sSR-A were 60.11% (107/178) and 63.35% (159/251), respectively. In early RA patients with disease duration less than 6 months, sSR-A also showed a positive rate of 53.85% (49/91). These results are described on Page 9-10 of the revised manuscript and shown in Figure 5A.

(3) PPV/NPV analysis of sSR-A in RA diagnosis: Following the suggestions of the Reviewer #1 and Reviewer #2, we further confirmed all clinical characteristics of RA patients as well as the disease controls, including SLE and SS patients, and provided the detailed information in the revised manuscript. Accordingly, we re-analyzed the sensitivity and specificity of sSR-A in RA diagnosis as well as the PPV/NPV. Based on the positive cut-off value (3SD above the mean value of the healthy controls), the sensitivity, specificity, PPV, and NPV of sSR-A for discrimination of RA in the test and validation cohorts were as follows:

- Beijing cohort (test cohort): sensitivity 61.36%, specificity 94.38%, PPV 86.17%, NPV 81.08%;
- Inner Mongolia cohort (validation cohort 1): sensitivity 73.24%, specificity 90.51%, PPV 75.73%, NPV 89.33%;
- Hangzhou cohort (validation cohort 2): sensitivity 74.19%, specificity 83.15%, PPV 71.88%, NPV 84.73%;
- Pooled three cohorts: sensitivity 66.41%, specificity 91.45%, PPV 80.19%, NPV 83.94%.

Since the sample size is relatively small in Inner Mongolia and Hangzhou cohorts, we pooled the results of the three cohorts, which is expected to be more accurate. The corresponding sensitivity of 66.41%, specificity of 91.45%, PPV of 80.19%, and NPV of 83.94% strongly support the powerful diagnostic value of sSR-A for RA. These results are described on Page 8-9 of the revised manuscript and shown in Table 1.

(4) Predictive value of sSR-A in undifferentiated arthritis (UA): Following the suggestions of the Reviewer #1 and Reviewer #2, we also examined the value of sSR-A as a predictor of RA. Serum samples from 119 UA patients were further collected for sSR-A detection. Although lower than those in ERA and RA patients, the levels of sSR-A were moderately increased in UA patients as compared with healthy controls. The prevalence of sSR-A in UA patients was 15.97% (19/119), comparable with anti-CCP (14.41%, 16/111) and RF (9.82%, 11/112). Moreover, those UA patients with high levels of sSR-A tended to display increased ESR or CRP, and positive RF. Studies of UA, ERA and RA patients showed that the levels and/or positive rates of sSR-A were increased gradually during disease progression. These findings indicate a potential value of sSR-A as a predictor of RA. Yet follow-up studies are still needed to further validate the predictive value, which will be performed in our future work. These results are described on Page 10-11 of the revised manuscript and shown in Figure 5C-D.

2. The levels of SR-A decline with treatment suggesting the possibility that the biomarker could be a disease activity biomarker. However, it is not indicated to what degree the

biomarker enhances the performance of currently used outcomes for measuring disease activity e.g. CRP.

Response:

As suggested, the performance of sSR-A was compared with ESR and CRP. RA patients were divided into the following four groups, and the levels of sSR-A as well as the positive rates were analyzed: (1) RA patients with normal ESR and normal CRP, (2) RA patients with normal ESR and increased CRP, (3) RA patients with increased ESR and normal CRP, and (4) RA patients with increased ESR and increased CRP.

The results showed that sSR-A demonstrated high prevalence in all these four groups, with differentially elevated levels. Even in RA patients with normal ESR and normal CRP, the positive rate of sSR-A still reached 57.58% (57/99). All these results indicate that sSR-A provides a complementary value to ESR and CRP. These results are described on Page 9 of the revised manuscript and shown in Figure 4.

3. Mice that receive the recombinant protein experience earlier disease onset and enhanced disease severity and more destruction of bone as measured by micro-CT. This was associated with elevated levels of IL17A although abrogation of the effect of SR-A with anti-IL17A was not tested. SR-A deficient mice on a C57BL/6 background develop less severe disease. Lack of SR-A also reduced the levels of TNF- α -producing CD4⁺ T cells.

Response:

Following the suggestions of the Reviewer #1 and Reviewer #2, we further investigated the relationship between IL-17A and sSR-A by performing IL-17A neutralization in wild-type (WT) mice with collagen-induced arthritis (CIA). We showed that IL-17A neutralization substantially ameliorated the disease severity in WT mice (**Supplementary Figure 7A**), but not to the extent with SR-A ablation in SRA^{-/-} mice (highly resistant to CIA). Although IL-17A neutralization alleviated the disease severity of CIA, it didn't reduce the levels of serum sSR-A (**Supplementary Figure 7B**). Additionally, incubation of bone marrow-derived macrophages and bone marrow-derived dendritic cells, both of which highly express SR-A on cell surface, with IL-17A didn't induce detectable secretion of sSR-A into the culture medium (**Supplementary Figure 8**). These results suggest that IL-17A may not directly trigger the cellular release of sSR-A despite that sSR-A represents a pathogenic factor capable of promoting disease progression and amplifying inflammatory response in RA. It is possible that production of sSR-A may require additional signals and occur prior to the elevation of IL-17A, and amplification of a Th17 response is only one of the mechanisms underlying the pathogenic role of sSR-A. These results are described on Page 15 of the revised manuscript and shown in Supplementary Figure 7 and 8. The discussion on the relationship between IL-17A and sSR-A is also provided on Page 18-19.

4. The severity of CIA was reduced with SR-A antibody mediated blockade.

Overall, the manuscript provides compelling evidence for a role of this mediator in the development of CIA but provides less compelling data for the value of this biomarker for diagnosis. The sample size for early RA and anti-CCP and/or RF negative RA is quite low. The PPV falls below 80% which would be considered a reasonable cut-off for assigning significant clinical utility. The data primarily emanates from well-established disease cohorts and the value of the test in patients presenting with arthralgia is unclear. It would be interesting to examine the value of this biomarker as a prognostic indicator. It would have helped the manuscript enormously if the authors had provided data from a longitudinal cohort attesting to its value as a prognostic indicator, especially for radiographic damage. That is where I think there is the most important unmet clinical need in the field, a modifiable prognostic biomarker. The data in CIA presented in this work suggests there could be a key role for SR-A as a prognostic biomarker.

Response:

In the revised manuscript, more data are provided to support the diagnostic value of soluble SR-A (sSR-A) in RA:

- (1) Diagnostic value of sSR-A in early RA (ERA): As described above, we have enlarged the sample size of ERA patients. Data from 251 ERA patients with disease duration less than 24 months, 178 ERA patients with disease duration less than 12 months, and 91 ERA patients with disease duration less than 6 months demonstrated that sSR-A revealed powerful diagnostic value in ERA. The positive rates of sSR-A in these three ERA groups were 63.35% (159/251), 60.11% (107/178), and 53.85% (49/91), respectively. Please refer to Page 9-10 and Figure 5A of the revised manuscript.
- (2) Diagnostic value of sSR-A in anti-CCP and/or RF negative RA: As described above, we have enlarged the sample size of anti-CCP and/or RF negative RA patients. Data from 179 anti-CCP-negative RA patients, 276 RF-negative RA patients, and 155 (anti-CCP+RF)-double negative RA patients support the use of sSR-A to complement the diagnosis in anti-CCP and/or RF negative RA patients. The positive rates of sSR-A were 49.72% (89/179) in anti-CCP-negative RA, 39.13% (108/276) in RF-negative RA, and 42.58% (66/155) in (anti-CCP+RF)-double negative RA. Please refer to Page 10 and Figure 5B of the revised manuscript.
- (3) PPV/NPV of sSR-A in RA diagnosis: As described above, pooled data from the test cohort and the two validation cohorts showed that the sensitivity and specificity of sSR-A for RA diagnosis were 66.41% and 91.45%, respectively. Moreover, the PPV was 80.19% and the NPV was 83.94%, supporting potential clinical utility. Please refer to Page 8-9 and Table 1 of the revised manuscript.
- (4) Complementary value of sSR-A to ESR and CRP: As described above, sSR-A displayed elevation with high prevalence in RA patients with normal ESR and/or normal CRP. Even in RA patients with normal ESR and normal CRP, the positive rate of sSR-A still reached 57.58% (57/99). These results indicate that sSR-A provides a complementary value to ESR and CRP. Please refer to Page 9 and Figure 4 of the revised manuscript.
- (5) Predictive value of sSR-A in undifferentiated arthritis (UA): As described above, our data from 119 UA patients demonstrated that, although lower than those in ERA and RA

patients, the levels of sSR-A were moderately increased in UA patients as compared with healthy controls. There was a 15.97% (19/119) prevalence of sSR-A in UA patients, which was comparable to anti-CCP (14.41%, 16/111) and RF (9.82%, 11/112). Moreover, those UA patients with high levels of sSR-A showed increased ESR or CRP, and positive RF. Comparing UA, ERA and RA patients revealed that the levels and/or positive rates of sSR-A were increased gradually during disease progression. While our findings support the potential value of sSR-A as a predictor of RA, follow-up studies are needed for validation. Please refer to Page 10-11 and Figure 5C-D of the revised manuscript.

- (6) Correlation of sSR-A with RA radiographic damage: The radiographic damage was assessed in 237 RA patients by the Sharp/van der Heijde score (SHS), and their correlation with sSR-A levels was further analyzed. The results showed that there was a modest correlation between sSR-A levels and RA patient radiographic damage. Moreover, sSR-A-positive RA patients showed relatively higher SHS than sSR-A-negative RA patients. These results suggest the value of sSR-A as a potential prognostic indicator in RA radiographic damage. However, follow-up studies are needed in our future work to further verify sSR-A as a prognostic indicator. Please refer to Page 12 and Supplementary Figure 1 of the revised manuscript.

Reviewer #2:

Hu F., et al measured the serum soluble scavenger receptor A in a large-scale test and validation cohorts and assessed the usefulness as a potential biomarker to support a diagnosis of rheumatoid arthritis. In addition, they tested the role of scavenger receptor A in collagen-induced arthritis model. Although the topics is interesting, there are a list of concerns below.

1. What is the rationale to identify this biomarker as a diagnostic biomarker? How did the authors find this biomarker as one of the best biomarkers among a long list of potential biomarkers? How did the authors collaborate that this biomarker is superior to many other biomarkers reported so far?

Response:

Rationale for choosing sSR-A as a potential RA diagnostic biomarker: The data presented in our manuscript were derived from a hypothesis-driven study. During the past two decades, we have been studying an immune modulating role of SR-A in various disease models, including cancer and autoimmune hepatitis. Our previous work documented that both cell-associated SR-A (cSR-A) and soluble SR-A (sSR-A) exhibit T cell suppressive activity via functional regulation of innate immune cells. However, the significance of sSR-A as a potential diagnostic marker in RA has not been investigated. Inspired by our previous studies, we hypothesized that sSR-A is present in patients with RA and has a diagnostic value in addition to its well-established role in immune modulation.

Although sSR-A exhibits T cell suppressive activity in disease models such as acute hepatitis, our current study reveals an unexpected effect of sSR-A in promoting Th17 cell activation in RA, which is characterized by chronic inflammation and bone erosion. Our findings are highly novel in that sSR-A plays pleiotropic roles in tissue inflammation, which we believe depend on the environmental cues and disease contexts. Based on the comments of the reviewer, we have added a description of the rationale for choosing sSR-A as a potential RA diagnostic biomarker in the INTRODUCTION SECTION of the revised manuscript (Page 6).

Superiority of sSR-A as compared with other biomarkers reported so far: As described in the revised manuscript, compared with other biomarkers reported so far, sSR-A demonstrates several superior features as follows:

- (1) sSR-A demonstrates powerful ability in discriminating RA. Our data of 2,679 serum samples from one test cohort and two validation cohorts showed that the sensitivity and specificity of sSR-A for RA diagnosis were 66.41% and 91.45%, respectively. Moreover, the PPV was 80.19% and the NPV was 83.94%, revealing potential clinical utility (**Figure 2, Table 1**).
- (2) sSR-A shows powerful diagnostic value in early RA (ERA). Our data from 251 ERA patients with disease duration less than 24 months, 178 ERA patients with disease duration less than 12 months, and 91 ERA patients with disease duration less than 6 months demonstrated that sSR-A revealed powerful diagnostic value in ERA. The positive rates of sSR-A in these three ERA groups were 63.35% (159/251), 60.11% (107/178), and 53.85% (49/91), respectively (**Figure 5A**).
- (3) sSR-A demonstrates potential for complementing the diagnosis in anti-CCP and/or RF negative RA. Our data from 179 anti-CCP-negative RA patients, 276 RF-negative RA patients, and 155 (anti-CCP+RF)-double negative RA patients showed that sSR-A demonstrated potential for complementing the diagnosis in anti-CCP and/or RF negative RA. The positive rates of sSR-A were 49.72% (89/179) in anti-CCP-negative RA, 39.13% (108/276) in RF-negative RA, and 42.58% (66/155) in (anti-CCP+RF)-double negative RA (**Figure 5B**).
- (4) sSR-A reveals complementary value to ESR and CRP. Our data from 253 RA patients with normal ESR and/or normal CRP showed that sSR-A demonstrated high prevalence in these RA patients with differentially elevated levels. Even in RA patients with normal ESR and normal CRP, the positive rate of sSR-A still reached 57.58% (57/99) (**Figure 4**).
- (5) sSR-A demonstrates potential predictive value in undifferentiated arthritis (UA). Our data from 119 UA patients showed that, although lower than those in ERA and RA patients, the levels of sSR-A were moderately increased in UA patients as compared with healthy controls. The prevalence of sSR-A in UA patients was 15.97% (19/119), comparable with anti-CCP (14.41%, 16/111) and RF (9.82%, 11/112). Moreover, those UA patients with high levels of sSR-A displayed increased ESR or CRP, and positive RF. Comparison of UA, ERA and RA patients showed that the levels and/or positive rates of sSR-A were increased during disease progression. These findings support the potential value of sSR-A as a predictor of RA (**Figure 5C-D**).

(6) sSR-A modestly correlates with RA radiographic damage. The radiographic damage was assessed in 237 RA patients by the Sharp/van der Heijde score (SHS), and their correlation with sSR-A levels was further analyzed. The results showed that there was a modest correlation between sSR-A levels and RA patient radiographic damage. Moreover, sSR-A-positive RA patients showed relatively higher SHS than sSR-A-negative RA patients. These results support the value of sSR-A as a potential prognostic indicator in RA radiographic damage, although follow-up studies are needed in our future work for validation (**Supplementary Figure 1**).

2. The authors should provide sufficient information about the treatment on RA and clinical characteristics of disease controls.

Response:

In the revised manuscript, we have provided the information regarding the treatment of RA patients in **Supplementary Table 3**. In addition, the clinical and demographic characteristics of the SLE patients (**Supplementary Table 4**), SS patients (**Supplementary Table 5**), as well as the early RA patients (**Supplementary Table 6**), anti-CCP and/or RF negative RA patients (**Supplementary Table 7**), and undifferentiated arthritis (UA) patients (**Supplementary Table 8**) have also been provided. Please refer to Page 57-62 of the revised manuscript.

3. In ACR/EULAR classification criteria, the rationale and golden standard to generate criteria is, first, to identify clinical and laboratory variables to predict initiation of DMARDs in undifferentiated arthritis, and then the potential variables are tested for the relative contribution in influencing the probability of developing 'persistent synovitis and/or erosive arthritis' as developing RA by using real-life case scenarios. Given these work flows, are the patients cohort used in this study appropriate to identify the new biomarkers to help early diagnosis of RA? In order to confirm the performance of biomarkers to early diagnosis, the authors should test the potential biomarkers in undifferentiated arthritis cohort and show the comparable or superior performance to the pre-existing laboratory variables.

Response:

Following the suggestions, we have also examined the performance of sSR-A in undifferentiated arthritis (UA). Serum samples from 119 UA patients were collected for sSR-A analysis. Although lower than those in ERA and RA patients, the levels of sSR-A moderately were increased in UA patients as compared with healthy controls. The prevalence of sSR-A in UA patients was 15.97% (19/119), comparable with anti-CCP (14.41%, 16/111) and RF (9.82%, 11/112). Moreover, those UA patients with high levels of sSR-A showed increased ESR or CRP, and positive RF. Analyses of UA, ERA and RA patients revealed that the levels and/or positive rates of sSR-A were increased during disease progression. While our findings support the potential value of sSR-A as a predictor of RA, follow-up studies are needed for validation. The results were described on Page 10-11 of the revised manuscript and demonstrated in Figure 5C-D.

4. In ACR/EULAR 2010 classification criteria, one has to exclude other diseases that best explained the signs and symptoms. In this regard, did the authors include appropriate disease controls to test specificity of the potential biomarkers including diseases with higher levels of inflammation such as polymyalgia rheumatic, adult onset still's disease, vasculitis and so on?

Response:

Based on the suggestions, we further examined the levels of sSR-A in those rheumatic diseases with higher levels of inflammation, including polymyalgia rheumatic (PMR), adult onset still's disease (AOSD), and ANCA-associated vasculitis (AAV). The results showed that compared with healthy controls, the serum levels of sSR-A were significantly elevated in RA patients, but not in patients with PMR, AOSD, or AAV, which further supports the specificity of sSR-A as a potential RA diagnostic biomarker. The results are described on Page 7 of the revised manuscript and shown in Figure 1A.

5. In ACR/EULAR 2010 classification criteria, CRP and ESR are listed in addition to anti-CCP and RF. Is the soluble scavenger receptor A superior to CRP and ESR? How did weigh this potential biomarker in the classification criteria?

Response:

As suggested, the performance of sSR-A was compared with ESR and CRP. RA patients were divided into the following four groups, and the levels of sSR-A as well as the positive rates were analyzed: (1) RA patients with normal ESR and normal CRP, (2) RA patients with normal ESR and increased CRP, (3) RA patients with increased ESR and normal CRP, and (4) RA patients with increased ESR and increased CRP.

The results showed that sSR-A demonstrated high prevalence in all these four groups, with differentially elevated levels. Even in RA patients with normal ESR and normal CRP, the positive rate of sSR-A still reached 57.58% (57/99). All these results indicate that sSR-A provides a complementary value to ESR and CRP. These results are described on Page 9 of the revised manuscript and shown in Figure 4.

6. Is the correlation between soluble SR-A and other clinical variables true for the inception cohort? The authors should demonstrate the correlation not only by cross-sectional cohorts, but also by longitudinal cohorts consisting of appropriate and sufficient numbers of the patients with responders and non-responders.

Response:

To further confirm the correlation of sSR-A with clinical and laboratory features in RA patients, we assessed the levels of sSR-A in both 100 non-responders ($DAS28 > 5.1$) and 54 responders ($DAS28 < 2.6$) of RA patients after therapy, and analyzed their clinical correlations, respectively. The results showed that the levels of sSR-A were significantly decreased in the responders but not in the non-responders of RA patients after therapy

(**Supplementary Figure 2**). Moreover, these correlations between sSR-A and clinical and laboratory variables in RA patients, as revealed in the inception cohorts (especially IgM, RF, and GPI), were more evident in the non-responders, yet couldn't be seen in the responders (**Supplementary Table 2**). The results are described on Page 12 of the revised manuscript and presented in Supplementary Figure 2 and Supplementary Table 2.

7. What is the biological role of soluble scavenger receptor on the pathogenesis of rheumatoid arthritis? Is this molecule mediating the persistent inflammation in synovium and involving in the joint destruction? Is this molecule regulated by inflammatory cytokines such as TNF-alpha and/ or IL-6?

Response:

Our current study has identified an intriguing pathogenic role of sSR-A in RA, including exacerbation of inflammation and bone erosion. This is directly supported by our observations that administration of SR-A protein to CIA mice resulted in more severe synovial inflammation and bone destruction (**Figure 6**), whereas blockade of SR-A in CIA mice using either SR-A-neutralizing antibody or SR-A inhibitor Fucoidan significantly alleviated the joint inflammation and bone destruction (**Figure 9**). However, additional studies are necessary for understanding precisely the role of sSR-A in the pathogenesis of RA.

Our new studies showed that stimulation of macrophages and dendritic cells, both of which highly express SR-A, with inflammatory cytokines (i.e., IL-1 β , IL-6, IL-17A, and TNF- α) failed to induce detectable secretion of sSR-A (**Supplementary Figure 8**). This suggests that elevation of sSR-A in RA may not be directly triggered by inflammatory cytokines or require additional signals. Additionally, the molecular size of sSR-A in the circulation is smaller than cell-associated SR-A (i.e., cSR-A, unpublished data) and various matrix metalloproteinases (e.g., MMP-3, MMP-9, and ADAM-17) are up-regulated in RA patients, suggesting that sSR-A may be derived from the cleavage of cSR-A. However, the molecular mechanisms underlying the production of sSR-A in RA remain to be determined. The results are described on Page 12-13 and 15-16 of the revised manuscript, discussed on Page 18-19, and shown in Figure 6, Figure 9, and Supplementary Figure 8.

8. In collagen-induced arthritis model without anti-citrullinated protein antibodies, are the levels of soluble scavenger receptor A correlating with the joint destruction, in addition to joint counts? Is the inhibition of SR-A ameliorating joint destruction? By blocking with anti-IL17A monoclonal antibodies, are the arthritis and joint destruction comparable to those inhibited by SR-A knock out? Are levels of soluble SR-A down-regulated by IL-17A? Given the apparent difference between human RA and CIA for the role of IL-17A, along with the indispensable role of IL-17 A in ankylosing spondylitis, the authors should comment on the role of SR-A in human RA and AS.

Response:

The clinical scores in CIA mice, assessed by joint swelling, typically correlate with the joint inflammation and destruction. As shown in **Figure 7A**, the sSR-A elevated along with increased arthritic scores in CIA mice, suggesting a positive correlation between the sSR-A levels and joint destruction. Furthermore, we showed that blockade of SR-A in CIA mice with either SR-A-neutralizing antibody or the SR-A inhibitor Fucoidan significantly mitigated the joint destruction as assessed by micro-CT (**Figure 9D**). All these results support the contribution of sSR-A to exacerbated joint destruction in RA. The results are described on Page 13-16 of the revised manuscript and shown in Figure 7 and Figure 9.

Based on the suggestions, we performed IL-17A neutralization in WT CIA mice to investigate the relationship between IL-17A and sSR-A. We showed that IL-17A neutralization ameliorated the disease severity in WT mice (**Supplementary Figure 7A**). Intriguingly, although IL-17A neutralization alleviated the disease severity, it didn't reduce the levels of serum sSR-A (**Supplementary Figure 7B**). Moreover, IL-17A was unable to induce the production of sSR-A by cultured myeloid cells (**Supplementary Figure 8**). Together, our data suggest that the cellular release of sSR-A requires additional signals present in RA and that elevation of sSR-A may occur prior to amplified Th17 response and IL-17A production. These results are described on Page 15 of the revised manuscript, discussed on Page 18-19, and shown in Supplementary Figure 7 and Supplementary Figure 8.

Our findings described above also partially explain why sSR-A isn't increased in AS patients in which IL-17A plays an indispensable role. Nevertheless, the involvement of SR-A in AS pathogenesis needs to be further investigated. We have discussed the potential role of SR-A in RA and AS patients in the revised manuscript (Page 19).

9. There are so many inconsistencies for the relationship between soluble SR-A and RF and the positive and negative role of soluble form of SR-A on T cell function. The authors should clarify the functional role of soluble form and membrane bound form of SR-A.

Response:

We appreciate the invaluable comment of the reviewer on the positive and negative roles of soluble SR-A (sSR-A) in immune regulation. While sSR-A suppresses T cell activation in disease model such as hepatitis, it promotes tissue inflammation and bone erosion in RA, as documented in the present study. It is conceivable that sSR-A have diverse and distinct biological or immunological effects in various inflammatory disorders, which may be governed by the environmental cues and disease contexts. Additional mechanistic studies are also necessary to identify those factors that function as sSR-A ligands in the context of inflammation, which may provide insights into functional diversity of sSR-A.

We have further clarified the functional roles of soluble form and membrane bound form of SR-A in the revised manuscript. Membrane bound form of SR-A was described as cell-associated SR-A (cSR-A), while soluble form of SR-A was described as sSR-A throughout the revised manuscript. Despite that both of these forms of SR-A exhibit T cell suppressive activity and contribute to tissue homeostasis in a model of hepatitis, sSR-A clearly demonstrates a role for promoting tissue inflammation in RA. We speculate that sSR-A in RA may compete with immunosuppressive cSR-A for immunoregulatory signals or

SR-A ligands present in the local environment, which causes functional impairment of cSR-A and dysregulated inflammatory response. Another possibility is that these sSR-A may interfere with the scavenging function of cSR-A or other scavenger receptors on cell surface, thereby disrupting their regulation of immune tolerance. However, a direct role of sSR-A in potentiating RA-associated inflammation cannot be excluded. A more detailed discussion has been provided in the revised manuscript (Page 18-19).

Reviewers' comments:

Reviewer #1 (Remarks to the Author):

The authors have provided significant additional cross-sectional data but no longitudinal data. Moreover, the cross-sectional data appears to be derived from a convenience sampling rather than consecutive patients with early RA. Consequently, there should be a reduction in the space devoted in Discussion to putative role of this biomarker in pathophysiology and a new section that describes study limitations. Without prospective data from consecutive patients with undiagnosed joint pain, the use of terms such as "powerful diagnostic biomarker" should be avoided. The performance characteristics of this biomarker are similar to anti-CCP and the main new finding is its presence in about 40% of patients who are RF and anti-CCP negative. Moreover, there are no data presented for patients with psoriatic arthritis.

Reviewer #2 (Remarks to the Author):

The authors extensively revised the manuscript according to the reviewers and provide new data support the revisions. The reviewer judge it appropriate for accept the revised manuscript.

Reviewer #3 (Remarks to the Author):

The paper entitled "Scavenger Receptor-A: Diagnostic Biomarker and Disease Exacerbator of 2 Rheumatoid Arthritis" identified and discussed about new biomarker sSR-A to classify the RA patient. This is a good piece work which has novel development and have potentiality in the field of medicine. However, I have some feedback on the statistical analysis of the paper, which are given below:

- i) The organization of section of the paper, where the method section is given in bottom of the paper. According to the standard setting, method section should be moved before the results and after the introduction
- ii) Need to describe the estimation of ROC curve and its area. There are three different approaches are available in literature: non-parametric, semiparametric, and parametric. It is not clear which approach they used.
- iii) There are some demographic and clinical covariates available in the data, which may be associated with both the biomarker and the disease. Need to investigate to identify any confounder that modify the association between the marker sSR-A the disease outcome, before starting to evaluate the performance of the marker in classifying RA patients from the healthy control. Multivariable logistic regression analysis might help to solve the problem.
- iv) No justification of naming test and validation data. Usually, researcher define training (where model is developed) and test data (or validation data- where predictive performance is assessed), where test and validation terms are the same.
- v) It seems that new marker sSR-A performed well in predicting RA patient. But need to compare their predictive performance with the existing markers. Need to show the ROC results in the same co-ordinate plot to distinguish how the new marker performed in comparison with the existing marker. Also need to show the sensitivity/specificity of both new and old marker for the dataset they used, if data on old marker is available. If data are not available, need to mentioned the performance of existing marker with reported ROC area and sensitivity/specificity.
- vi) When new marker is identified with reasonably better predictive performance, it is necessary to determine the optimal cutoff value for the maker, which can also be done using ROC analysis and Youden index etc.

Detailed responses to the reviewers' comments:

We appreciate the valuable suggestions of the expert reviewers. We have conducted additional experiments and included new data in the revised manuscript. The underlying statistical analyses in the study have been checked throughout the manuscript. The point-to-point responses to each question are as follows:

Reviewer #1:

The authors have provided significant additional cross-sectional data but no longitudinal data. Moreover, the cross-sectional data appears to be derived from a convenience sampling rather than consecutive patients with early RA. Consequently, there should be a reduction in the space devoted in Discussion to putative role of this biomarker in pathophysiology and a new section that describes study limitations. Without prospective data from consecutive patients with undiagnosed joint pain, the use of terms such as "powerful diagnostic biomarker" should be avoided. The performance characteristics of this biomarker are similar to anti-CCP and the main new finding is its presence in about 40% of patients who are RF and anti-CCP negative. Moreover, there are no data presented for patients with psoriatic arthritis.

Response:

Many thanks for the suggestions. The following changes have been made in the revised manuscript:

- (1) In the Discussion Section, we have reduced the discussion on the putative role of SR-A in RA pathogenesis. The study limitations have been described in details (page 17, paragraph 2).
- (2) The description terms have been revised throughout the manuscript. "Powerful diagnostic biomarker" has been changed to "potential diagnostic biomarker".
- (3) The levels of sSR-A in psoriatic arthritis (PsA) patients have been examined. Compared with healthy controls, the serum levels of sSR-A were significantly elevated in RA

patients, but not in patients with PsA. The results are described on Page 6 of the revised manuscript (Figure 1a).

Reviewer #2:

The authors extensively revised the manuscript according to the reviewers and provided new data support the revisions. The reviewer judge it appropriate for accept the revised manuscript.

Response:

Thanks so much for the supportive comment on the manuscript. We are very much appreciated.

Reviewer #3:

The paper entitled “Scavenger Receptor-A: Diagnostic Biomarker and Disease Exacerbator of Rheumatoid Arthritis” identified and discussed about new biomarker sSR-A to classify the RA patient. This is a good piece work which has novel development and has potentiality in the field of medicine. However, I have some feedback on the statistical analysis of the paper, which are given below:

i) The organization of section of the paper, where the method section is given in bottom of the paper. According to the standard setting, method section should be moved before the results and after the introduction.

Response:

Thank you for the suggestion. The section organization of the manuscript complies with the requirements of *Nature Communications*.

ii) Need to describe the estimation of ROC curve and its area. There are three different approaches available in literature: non-parametric, semiparametric, and parametric. It is not clear which approach they used.

Response:

As suggested, the estimation of ROC curve, its area and the method used have been added in the revised manuscript.

In this study, multivariable logistic regression analysis and ROC curve analysis with non-parametric method were used to evaluate the performance of sSR-A in diagnosis of RA, with age and gender as the confounders. The results revealed a significant area under the curve (AUC) of 0.902 (95% CI extending from 0.883 to 0.922) for sSR-A in Beijing cohort (training cohort). The AUC of Inner Mongolia cohort (validation cohort 1) and Hangzhou cohort (validation cohort 2) were 0.914 (95% CI extending from 0.887 to 0.941) and 0.913 (95% CI extending from 0.886 to 0.941), respectively. Pooling the data of the three cohorts yielded an AUC of 0.906 (95% CI extending from 0.892 to 0.920) for sSR-A. These results indicate that sSR-A has a potential capacity in distinguishing RA.

The above information is described in the Results Section (page 7, paragraph 2) and Methods Section (page 21, paragraph 3) of the revised manuscript and presented in Figure 3.

iii) There are some demographic and clinical covariates available in the data, which may be associated with both the biomarker and the disease. Need to investigate to identify any confounder that modify the association between the marker sSR-A the disease outcome, before starting to evaluate the performance of the marker in classifying RA patients from the healthy control. Multivariable logistic regression analysis might help to solve the problem.

Response:

As described above, multivariable logistic regression analysis followed by ROC curve analysis with non-parametric method was performed to evaluate the performance of sSR-A in

diagnosis of RA, with age and gender as the confounders. The analytic method used in the current version is consistent with the suggestions of the reviewer.

The clarifications have been made in the Results Section (page 7, paragraph 2) and Methods Section (page 21, paragraph 3) of the revised manuscript.

iv) No justification of naming test and validation data. Usually, researcher define training (where model is developed) and test data (or validation data- where predictive performance is assessed), where test and validation terms are the same.

Response:

As suggested, terms of training and validation data are defined and used in the revised manuscript.

v) It seems that new marker sSR-A performed well in predicting RA patient. But need to compare their predictive performance with the existing markers. Need to show the ROC results in the same co-ordinate plot to distinguish how the new marker performed in comparison with the existing marker. Also need to show the sensitivity/specificity of both new and old marker for the dataset they used, if data on old marker is available. If data are not available, need to mention the performance of existing marker with reported ROC area and sensitivity/specificity.

Response:

The reported ROC area for anti-CCP and RF are 0.84 and 0.83, respectively, while the reported average sensitivity and specificity are 67% and 95% for anti-CCP, and 69% and 85% for RF, respectively (references attached below).

As suggested, these are described in the Results Section (page 7, paragraph 2 and page 8, paragraph 1) of the revised manuscript.

References in the revised manuscript:

11. Nishimura, K., et al. *Meta-analysis: diagnostic accuracy of anti-cyclic citrullinated peptide antibody and rheumatoid factor for rheumatoid arthritis. Annals of Internal Medicine* 146, 797-808 (2007).

26. Vallbracht, I., et al. *Diagnostic and clinical value of anti-cyclic citrullinated peptide antibodies compared with rheumatoid factor isotypes in rheumatoid arthritis. Ann Rheum Dis* 63, 1079-1084 (2004).

vi) *When new marker is identified with reasonably better predictive performance, it is necessary to determine the optimal cutoff value for the maker, which can also be done using ROC analysis and Youden index etc.*

Response:

Thanks for the insightful comment. As determined by ROC curve analysis with the highest Youden index in this study, the sensitivity and specificity of sSR-A in RA were 81.10% and 85.70%, respectively. The optimal cut-off value was set for 3 SD above the mean value of the healthy controls, which showed better clinical utility potentially than the ROC curve analysis, yielding the sensitivity of 66.41% and specificity of 91.45%.

These have been described in the Results Section (page 8, paragraph 1) and Methods Section (page 21, paragraph 3) of the revised manuscript.

REVIEWERS' COMMENTS:

Reviewer #3 (Remarks to the Author):

Thank you very much addressing my comments. I am still confused with one correction. You used multivariable logit model for adjusting for effect of other confounder when assessing the predictive accuracy of new biomarker. Actually there are some published papers where they discussed on covariate adjusted ROC curve for assessing classification accuracy of new biomarker. Only using Multivariable model to adjust for several covariates indicate the predictive performance of the model not solely the marker.

Detailed responses to the reviewers' comments:

We appreciate the valuable suggestion of the expert reviewer. Additional statistical analysis has been performed and included in the revised manuscript. Please see our point-to-point response below to the reviewer's comment.

Reviewer #3 (Remarks to the Author):

Thank you very much addressing my comments. I am still confused with one correction. You used multivariable logit model for adjusting for effect of other confounder when assessing the predictive accuracy of new biomarker. Actually there are some published papers where they discussed on covariate adjusted ROC curve for assessing classification accuracy of new biomarker. Only using Multivariable model to adjust for several covariates indicate the predictive performance of the model not solely the marker.

Response:

As suggested, we performed the covariate-adjusted receiver operating characteristic curve (AROC) analysis using non-parametric method to evaluate the performance of sSR-A in diagnosis of RA, with age and gender as the confounders identified by the multivariable logistic regression analysis. The AROC analysis was performed using the Stata programs developed by Dr. Pepe's group (see references below and more information at <https://research.fhcrc.org/diagnostic-biomarkers-center/en/software.html>).

The results showed a significant area under the age- and gender-adjusted ROC curve (AAUC) of 0.8420 (95% CI extending from 0.8094 to 0.8688) for sSR-A in Beijing cohort (training

cohort). The AAUCs of Inner Mongolia cohort (validation cohort 1) and Hangzhou cohort (validation cohort 2) were 0.8641 (95% CI extending from 0.8232 to 0.9013) and 0.8219 (95% CI extending from 0.7502 to 0.8748), respectively. Pooling the data of the three cohorts yielded an AAUC of 0.8436 (95% CI extending from 0.8225 to 0.8669) for sSR-A. All these results support the potential capacity of using sSR-A for disease diagnosis.

The above information has been included in the Results Section (page 8, line 17-page 9, line 13) and Methods Section (page 23, lines 4-16) of the revised manuscript with tracked changes and presented in Figure 3.

Please refer to page 7 and page 20 of the merged PDF file.

References included in the revised manuscript:

25. Janes, H. & Pepe, M.S. *Adjusting for covariates in studies of diagnostic, screening, or prognostic markers: an old concept in a new setting. Am J Epidemiol* 168, 89-97 (2008).
26. Janes, H., Longton, G. & Pepe, M. *Accommodating Covariates in ROC Analysis. Stata J* 9, 17-39 (2009).
27. Janes, H. & Pepe, M.S. *Adjusting for covariate effects on classification accuracy using the covariate-adjusted receiver operating characteristic curve. Biometrika* 96, 371-382 (2009).